# Phasor field diffraction based reconstruction for fast non-line-of-sight imaging systems

Xiaochun Liu [1], Sebastian Bauer [2] & Andreas Velten [1,2]✉

Non-line-of-sight (NLOS) imaging recovers objects using diffusely reflected indirect light using transient illumination devices in combination with a computational inverse method. While capture systems capable of collecting light from the entire NLOS relay surface can be much more light efficient than single pixel point scanning detection, current reconstruction algorithms for such systems have computational and memory requirements that prevent real-time NLOS imaging. Existing real-time demonstrations also use retroreflective targets and reconstruct at resolutions far below the hardware limits. Our method presented here enables the reconstruction of room-sized scenes from non-confocal, parallel multi-pixel measurements in seconds with less memory usage. We anticipate that our method will enable real-time NLOS imaging when used with emerging single-photon avalanche diode array detectors with resolution only limited by the temporal resolution of the sensor.

[1] Department of Electrical and Computer Engineering, University of Wisconsin – Madison, Madison, WI, USA. [2] Department of Biostatistics and Medical Informatics, University of Wisconsin – Madison, Madison, WI, USA. ✉email: velten@wisc.edu

Time of flight Non-line-of-sight (NLOS) imaging uses fast pulsed light sources and detectors combined with computational methods to image scenes from indirect light reflections making it possible to reconstruct images or geometry of the parts of a scene that are occluded from direct view. Due to this unique capability, NLOS imaging is promising for applications in diverse fields such as law enforcement, infrastructure assessment, flood prevention, border control, disaster response, planetary research, geology, volcanology, manufacturing, industrial monitoring, vehicle navigation, collision avoidance, and military intelligence. In a time-resolved NLOS imaging measurement, points on a relay wall are illuminated by a picosecond laser. Light from these points illuminates the hidden scene and a fast detector captures the optical signal returned from the scene at points on the relay wall. A suitable computational method is then used to decode the image around the corner.

Despite recent breakthroughs, obtaining a high resolution real time or near real time NLOS video remains elusive. An algorithm suitable for fast NLOS imaging must fulfill three separate requirements: The ability to use data that can be captured in real time, a computational complexity allowing for execution in a fraction of a second on a conventional CPU or GPU, and a memory complexity suitable for use in the limited memory of such a system.

After theoretical exploration of the problem[1,2], the first experimental demonstration of NLOS imaging used a filtered backprojection (FBP) algorithm[3,4] similar to inverse methods used in computed tomography. Modified FBP algorithms such as error backprojection[5] and Laplacian of Gaussian (LOG) FBP[6] can provide high quality reconstructions, but have a high computational complexity and take minutes to hours to execute on a desktop computer. Buttafava et al.[7] show that it is possible to use a gated Single-Photon Avalanche Diode (SPAD) for NLOS imaging. SPADs can potentially be manufactured at low cost and in large arrays enabling fast parallel NLOS capture.

Among the fastest current reconstruction methods, O'Toole et al.[8] propose a Light Cone Transform (LCT) method based on co-located illumination and detection points and acquire all measurements through a scanning process of the relay wall (so-called confocal acquisition setup). Lindell et al.[9] demonstrate another reconstruction method for confocal data transferred from seismic imaging which is called FK Migration. Both algorithms rely on 3D convolutions allowing for fast reconstruction and demonstrate the ability to recover complex scenes from confocal measurements[9]. They require interpolation over irregular 3D grids in order to approximate the data points needed for the convolutions. This requires oversampling the reconstructions and computing nearest neighbors which is associated with significant added memory requirements. The crucial limitation of these methods that we explore in more detail below is, however, that they can only utilize the light returning from the confocal location on the relay wall and thus cannot utilize the vast majority of light available in an NLOS measurement. This is illustrated in our Supplementary Note 4. Lindell et al. also demonstrate a way to approximate non-confocal data as confocal data[9] for simple planar scenes that allows both LCT and FK Migration algorithms to obtain approximate reconstructions from non-confocal data. Real time reconstruction of low resolution retro-reflective scenes has also been demonstrated in a confocal scanning scenario with both LCT and FK Migration methods. However, the presented confocal real time captures require retroreflective targets that return most reflected light to the moving laser/detection point, while arbitrary diffuse objects require scan times of at least 10 minutes[9]. In this case, the bottleneck of these methods is not the computation, but the acquisition. Furthermore, reconstruction of higher resolution scenes with diffuse surfaces is hindered by the large memory requirements and the inefficient confocal capture process requiring sequential point scanning capture with a single SPAD pixel.

Liu et al.[10] and Reza et al.[11] introduce a virtual wave phasor field formalism that is the basis of this work. Using the phasor field method, the NLOS imaging problem can be stated as a line of sight optical imaging problem based on diffraction and solved using existing diffraction theory methods. Recent work also includes further experimental investigation in the propagation of phasor field virtual waves[12], as well as the extension of the phasor field model to scenes with occlusions and specular reflectors[13] who use the paraxial approximation to obtain an approximate convolution operator to model wave propagation. More insight into the theory of phasor field waves is also provided by Teichman et al.[14].

In this work, we introduce an NLOS reconstruction method using the phasor field formalism along with a convolutional fast Fourier transform (FFT) based Rayleigh Sommerfeld Diffraction (RSD) algorithm to provide fast non-approximative scene reconstructions for general capture setups, in particular including non-confocal setups using a single laser and a sensor array. Our hardware prototype includes a SPAD detector and a picosecond pulse laser which will be mentioned specifically later. When used in the confocal scenario, this new method performs at speed similar to LCT and FK Migration, while requiring significantly less memory. In addition to applying our new algorithm to open source data[9,10], we also perform several additional experiments.

## Results

**Phasor field NLOS camera.** The concept of phasor field NLOS imaging is described in Fig. 1. Data from the scene is collected by illuminating a set of points $\mathbf{x}_p$ on a relay surface $P$ and collecting the light returned at points $\mathbf{x}_c$ on a relay surface $C$. This data set represents impulse responses $H(\mathbf{x}_p \rightarrow \mathbf{x}_c, t)$ of the scene. Using such an impulse response we can compute the scene response at points $\mathbf{x}_c$ to an input signal $\mathcal{P}(\mathbf{x}_p, t)$ as

$$\mathcal{P}(\mathbf{x}_c, t) = \int_P [\mathcal{P}(\mathbf{x}_p, t) \underset{t}{*} H(\mathbf{x}_p \rightarrow \mathbf{x}_c, t)] d\mathbf{x}_p \quad (1)$$

where the $\underset{t}{*}$ operator indicates a convolution in time. We call the quantities $\mathcal{P}(\mathbf{x}_p, t)$ and $\mathcal{P}(\mathbf{x}_c, t)$ phasor field wavefronts. $\mathcal{P}(\mathbf{x}_c, t)$ describes the wavefront that would be returned from the scene if it were illuminated by a illumination wave $\mathcal{P}(\mathbf{x}_p, t)$. Reconstructing an image from the wave front of a reflected wave is the fundamental problem solved by a line of sight imaging system. The reconstruction operation

$$I(\mathbf{x}_v, t) = \Phi(\mathcal{P}(\mathbf{x}_c, t)) \quad (2)$$

resulting in a 3D image $I(\mathbf{x}_v)$ of the scene amounts to propagation of the wavefront at $C$ back into the scene into the points $\mathbf{x}_v$ where it has the shape of the scene objects. The Fourier domain version $\Phi_{\mathcal{F}}(\cdot)$ of the wave propagation operator $\Phi(\cdot)$ is known as the Rayleigh-Sommerfeld Diffraction (RSD) integral:

$$\Phi(\mathcal{P}_{\mathcal{F}}(\mathbf{x}_c, \Omega)) = \left| \mathcal{R}_{\mathbf{x}_v}(\mathcal{P}_{\mathcal{F}}(\mathbf{x}_c, \Omega)) \right|^2. \quad (3)$$

The RSD in the considered context is calculated by

$$\mathcal{R}_{\mathbf{x}_v}(\mathcal{P}_{\mathcal{F}}(\mathbf{x}_c, \Omega)) = \alpha(\mathbf{x}_v) \int_C \mathcal{P}_{\mathcal{F}}(\mathbf{x}_c, \Omega) \underbrace{\frac{e^{-ik|\mathbf{x}_c - \mathbf{x}_v|}}{|\mathbf{x}_c - \mathbf{x}_v|}}_{\text{RSD diffraction kernel}} d\mathbf{x}_c.$$

$$(4)$$

In this equation, $k = \Omega/c$ denotes the wavenumber and $c$ across our paper refers to the speed of light. The conventional RSD

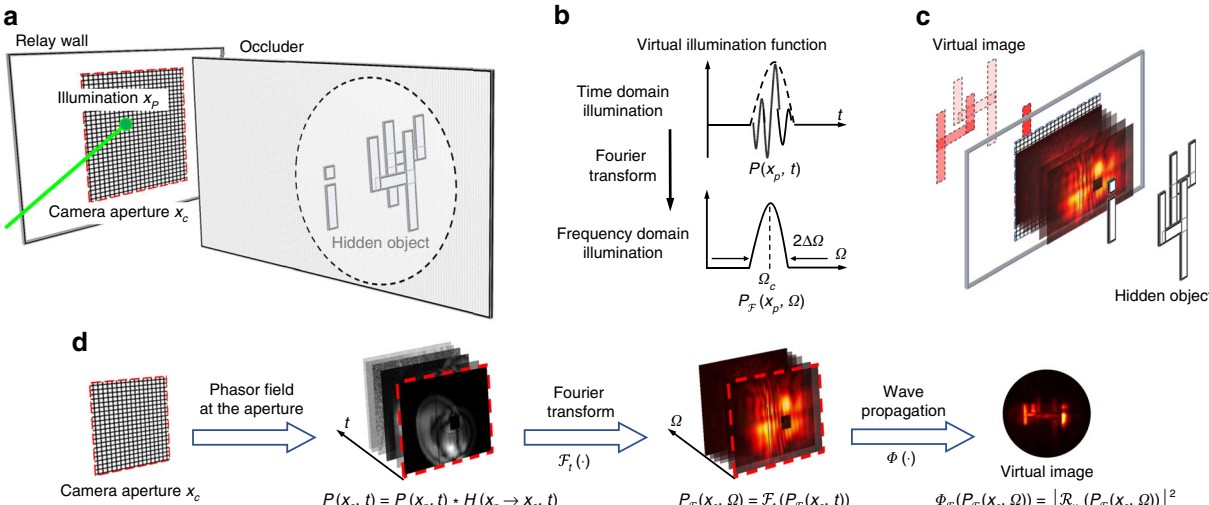

**Fig. 1 Illustration of the proposed fast phasor field NLOS imaging method. a** The NLOS imaging scenarios, including relay wall, occluder, and hidden object. Measurements are performed on the relay wall, including illumination point $\mathbf{x}_p$ and camera aperture $\mathbf{x}_c$. **b** The virtual illumination in the reconstruction in time and frequency domain. **d** The entire reconstruction pipeline. The wave propagation model is described in the following. Overall, our proposed method can be thought of as building a virtual lens as shown in **c**, which creates the corresponding virtual image of hidden objects from the captured phasor field.

propagates the electric field, but in this context propagation of an intensity modulation is required. The phasor field RSD differs from the conventional version by the amplitude correction factor $\alpha(\mathbf{x}_v)$[10]. This factor depends on the location $\mathbf{x}_v$ of the reconstruction point and could be precomputed once the geometry of the relay surface is known. Alternatively, it can be disregarded, as it only causes a slowly varying error in brightness of reconstructed points, but not their location. The RSD in Eq. (4) is a function of each individual monochromatic phasor field component. For this reason, the wavefront $\mathcal{P}(\mathbf{x}_c, t)$ received at the aperture has been replaced by its Fourier domain representation $\mathcal{P}_{\mathcal{F}}(\mathbf{x}_c, \Omega)$. Throughout this paper, frequency domain quantities are denoted by the same variable as the respective time domain quantities, but with the subscript $\mathcal{F}$ and the argument angular frequency $\Omega$ instead of $t$. For instance, $\mathcal{F}_t\big(\mathcal{P}(\mathbf{x}_p, t)\big) = \mathcal{P}_{\mathcal{F}}(\mathbf{x}_p, \Omega)$ and $\mathcal{F}_t\big(H(\mathbf{x}_p \to \mathbf{x}_c, t)\big) = H_{\mathcal{F}}(\mathbf{x}_p \to \mathbf{x}_c, \Omega)$, where $\mathcal{F}_t(\cdot)$ denotes the Fourier transform with respect to time. Note that in this paper, the RSD propagation direction is from the camera aperture (i.e., relay surface $C$) into the reconstruction volume.

It is important to note that both illumination $\mathcal{P}(\mathbf{x}_p, t)$ and image formation $\Phi(\cdot)$ are implemented virtually on a computer. For this reason, they can be chosen to mimic any LOS imaging system. For the purpose of NLOS 3D image reconstruction, one option is to choose a transient camera sending a virtual phasor field pulse

$$\mathcal{P}(\mathbf{x}_p, t) = e^{i\Omega_C t}\delta(\mathbf{x}_p - \mathbf{x}_{ls})e^{-\frac{(t-t_0)^2}{2\sigma^2}} \qquad (5)$$

from the virtual light source position $\mathbf{x}_{ls}$ into the scene. The center frequency $\Omega_C$ has to be chosen according to the spatial relay wall sampling. The smallest achievable wavelength should be larger than twice the largest distance between neighboring points $\mathbf{x}_p$ and $\mathbf{x}_c$ and larger than the temporal resolution of the imaging hardware[10]. For example, given a spatial sampling of 1 cm, the smallest possible modulation wavelength is larger than 2 cm. For the following, we set $t_0 = 0$. The illumination pulse as a function of time needs to be converted into the frequency domain, so that each corresponding frequency is then propagated separately by

the RSD in Eq. (4). The temporal Fourier transform of the illumination phasor field yields

$$\begin{aligned}
\mathcal{P}_{\mathcal{F}}(\mathbf{x}_p, \Omega) &= \mathcal{F}_t\big(\mathcal{P}(\mathbf{x}_p, t)\big) \\
&= \delta(\mathbf{x}_p - \mathbf{x}_{ls})\left(2\pi\delta(\Omega - \Omega_C) \underset{f}{*} \sigma\sqrt{2\pi}e^{-\frac{\sigma^2\Omega^2}{2}}\right).
\end{aligned}$$
$$(6)$$

The result $\mathcal{P}_{\mathcal{F}}(\mathbf{x}_p, \Omega)$ in the frequency domain is a Gaussian centered around the central frequency $\Omega_C$ as it is shown in Fig. 1. Figuratively, the RSD propagates the light wave arriving at the aperture (i.e., relay surface $C$) back into the scene, thereby reconstructing it. Equivalently, one can think of it as a virtual imaging system that forms the image acquired by a virtual sensor behind the relay wall.

After processing all frequency components through space with the RSD, the result at $\mathbf{x}_v$ needs to be converted to the time domain again by applying the inverse Fourier transform. The overall reconstruction is therefore calculated by

$$I(\mathbf{x}_v, t) = \left| \int_{-\infty}^{+\infty} e^{i\Omega t} \mathcal{R}_{\mathbf{x}_v}\left( \underbrace{\underbrace{\mathcal{P}_{\mathcal{F}}(\mathbf{x}_p, \Omega)}_{\text{Illumination phasor field}} \cdot H_{\mathcal{F}}(\mathbf{x}_p \to \mathbf{x}_c, \Omega)}_{\text{Phasor field at the camera aperture (relay surface } C)} \right) \frac{d\Omega}{2\pi} \right|^2,$$
$$(7)$$

where the integral over P has vanished as there is only one virtual illumination point $\mathbf{x}_{ls}$. Calculating the square is omitted in the actual reconstruction implementation, as it only affects the scene contrast.

**Fast phasor field diffraction.** The main goal of this paper is to develop a fast 3D NLOS reconstruction method based on the RSD propagator $\mathcal{R}_{\mathbf{x}_v}(\cdot)$ in Eq. (7). Multiple convolutional RSD methods have been introduced in the literature[15–17] and form the basis of our approach.

The RSD as defined in Eq. (4) can propagate the wave from an arbitrary surface to any arbitrary point $\mathbf{x}_v$. For fast

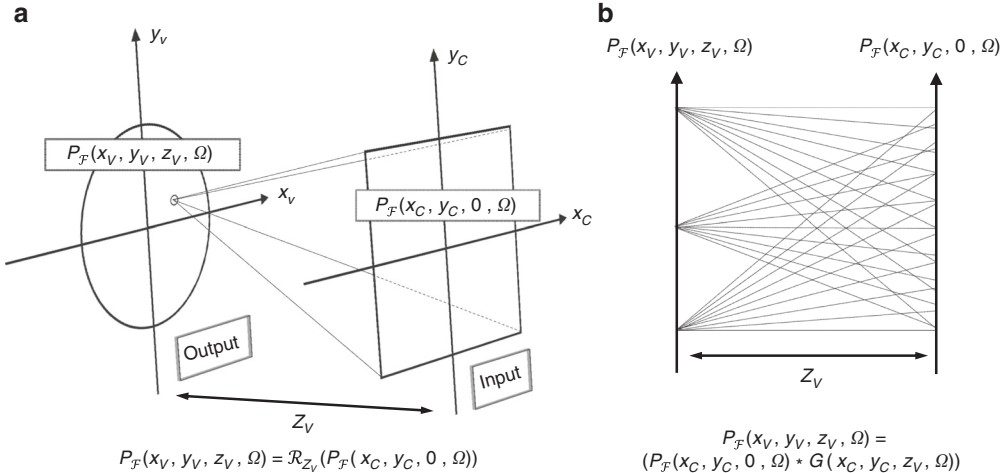

**Fig. 2 Rayleigh Sommerfeld Diffraction (RSD) calculation. a** Two parallel planes geometrical setup for the reconstruction. The input and output planes are space with $z_v$. **b** Side view for **a**.

implementation we constrain the operator to propagate the wave between two parallel planes. This allows us to work with two spatial dimensions. We introduce the scalar coordinates $\mathbf{x}_c = (x_c, y_c, 0)$ and $\mathbf{x}_v = (x_v, y_v, z_v)$ and rewrite the RSD in Eq. (4) as follows:

$$
\begin{aligned}
\mathcal{P}_{\mathcal{F}}(\mathbf{x}_v, \Omega) &= \mathcal{R}_{z_v}(\mathcal{P}_{\mathcal{F}}(\mathbf{x}_p, \Omega) \cdot H_{\mathcal{F}}(\mathbf{x}_p \rightarrow \mathbf{x}_c, \Omega)) \\
&= \mathcal{R}_{z_v}(\mathcal{P}_{\mathcal{F}}(\mathbf{x}_c, \Omega)) \\
\mathcal{P}_{\mathcal{F}}(x_v, y_v, z_v, \Omega) &= \mathcal{R}_{z_v}(\mathcal{P}_{\mathcal{F}}(x_c, y_c, 0, \Omega)) \\
&= \iint_{-\infty}^{+\infty} \mathcal{P}_{\mathcal{F}}(x_c, y_c, 0, \Omega) \underbrace{\frac{\alpha(x_v, y_v, z_v) e^{-\frac{\Omega}{c}\sqrt{(x_c - x_v)^2 + (y_c - y_v)^2 + z_v^2}}}{\sqrt{(x_c - x_v)^2 + (y_c - y_v)^2 + z_v^2}}}_{\text{RSD diffraction kernel}} \, dx_c \, dy_c \\
&= \iint_{-\infty}^{+\infty} \mathcal{P}_{\mathcal{F}}(x_c, y_c, 0, \Omega) \cdot \underbrace{G(x_v - x_c, y_v - y_c, z_v, \Omega)}_{\text{2D convolution kernel}} \, dx_c \, dy_c \\
&= \underbrace{\mathcal{P}_{\mathcal{F}}(x_c, y_c, 0, \Omega) * G(x_c, y_c, z_v, \Omega)}_{\text{Spatial 2D convolution}},
\end{aligned}
$$

(8)

where the geometrical setup is illustrated in Fig. 2. Equation (8) considers two parallel planes with spacing $z_v$ in a Cartesian coordinate system. For this reason, the RSD notation changed from $\mathcal{R}_{\mathbf{x}_v}(\cdot)$ for the point $\mathbf{x}_v$ to $\mathcal{R}_{z_v}(\cdot)$ to indicate that the propagation holds for all points in the plane at distance $z_v$ from the relay wall. For a single frequency component $\Omega$, the relation between the wavefront $\mathcal{P}_{\mathcal{F}}(x_c, y_c, 0, \Omega)$ at the camera aperture plane and the wavefront $\mathcal{P}_{\mathcal{F}}(x_v, y_v, z_v, \Omega)$ at the virtual image plane is a two-dimensional spatial convolution with the 2D convolution kernel defined by $G(x_c, y_c, z_v, \Omega) = \frac{\alpha(x_v, y_v, z_v) \cdot \exp(-i\frac{\Omega}{c}\sqrt{x_c^2 + y_c^2 + z_v^2})}{\sqrt{x_c^2 + y_c^2 + z_v^2}}$ where the factor $\alpha(x_v, y_v, z_v)$ will be ignored during reconstruction. Note that the RSD in Eq. (8) needs to be calculated for each individual frequency component $\mathcal{P}_{\mathcal{F}}(x_c, y_c, 0, \Omega)$. Considering the virtual pulse illumination in Eq. (5), the wavefront $\mathcal{P}_{\mathcal{F}}(x_c, y_c, 0, \Omega)$ is a broad-band signal; its spectrum is a Gaussian centered around $\Omega_C$ as shown in Eq. (6). For this reason, it is sufficient to consider the frequency range $\Omega \in [\Omega_C - \Delta\Omega, \ \Omega_C + \Delta\Omega]$. Although the magnitude is not completely zero outside this interval, it is very small and can be neglected. The chosen range $\Delta\Omega$ depends on the virtual illumination pulse bandwidth and thus on the pulse width parameter $\sigma$. Thus, applying Eq. (8) for the frequencies $\Omega \in [\Omega_C - \Delta\Omega, \ \Omega_C + \Delta\Omega]$ and subsequent inverse Fourier

transform with respect to time

$$
\mathcal{P}(x_v, y_v, z_v, t) = \int_{\Omega_C - \Delta\Omega}^{\Omega_C + \Delta\Omega} e^{i\Omega t} \cdot \underbrace{\mathcal{R}_{z_v}\left(\mathcal{P}_{\mathcal{F}}(x_c, y_c, 0, \Omega)\right)}_{\text{Monochromatic wavefront at depth } z_v} \frac{d\Omega}{2\pi} \quad (9)
$$

is equivalent to sending the designed modulated virtual illumination pulse wavefront $\mathcal{P}(\mathbf{x}_p, t) = e^{i\Omega_C t} e^{-\frac{t^2}{2\sigma^2}}$ into the hidden scene, capturing its reflection at the visible relay wall, and propagating it back into the scene or imaging it onto a virtual imaging sensor using a virtual lens. The relay wall functions as a virtual aperture. The output $\mathcal{P}(x_v, y_v, z_v, t)$ in Eq. (9) depends on the time $t$, as each reconstruction point is illuminated only for a short period of time. Taking the absolute value of $\mathcal{P}(x_v, y_v, z_v, t)$ in Eq. (9) and squaring it makes us arrive at a 4D reconstruction (cf. Eq. (7)). We can understand this reconstruction as a movie of a virtual pulse traveling through the hidden scene, as shown in Fig. 3. In this figure, a patch shaped as a 4 is being illuminated by a spherical wavefront coming from the illumination point on the relay surface.

The process of reconstructing a 4D model for achieving a 3D spatial reconstruction is unnecessarily time-consuming. The 3D reconstruction of the scene can be obtained from the movie by freezing the time of arrival corresponding to the peak of the illumination pulse for each voxel. Thus each voxel contains only the direct (3rd) bounce signal from the hidden object. This can be performed by calculating the spherical geometry as a function of point source illumination position $(x_{ls}, y_{ls}, 0)$ and replacing $t$ at each voxel $(x_v, y_v, z_v)$:

$$
t := \frac{1}{c}\sqrt{(x_v - x_{ls})^2 + (y_v - y_{ls})^2 + z_v^2}. \quad (10)
$$

In this equation, we have used the scalar representation $\mathbf{x}_{ls} = (x_{ls}, y_{ls}, 0)$ of the virtual light source position. Replacing $t$ by the appropriate spatial coordinates as described in this equation leads to a 3D virtual camera that only sees the direct bounce from the hidden object and removes the fourth dimension; the respective voxels are reconstructed at exactly the time when the pulse arrives. This leads to a more time-efficient reconstruction than acquiring the full 4D wavefront.

However, there is one problem that needs to be taken care of. While the RSD in Eq. (8) is calculated at planes parallel to the aperture, the illumination pulse spreads spherically from the light source. Theoretically, this means each point on the plane should be reconstructed with a different time shift as given in Eq. (10) which leads to an integral expression, i.e., the inverse Fourier

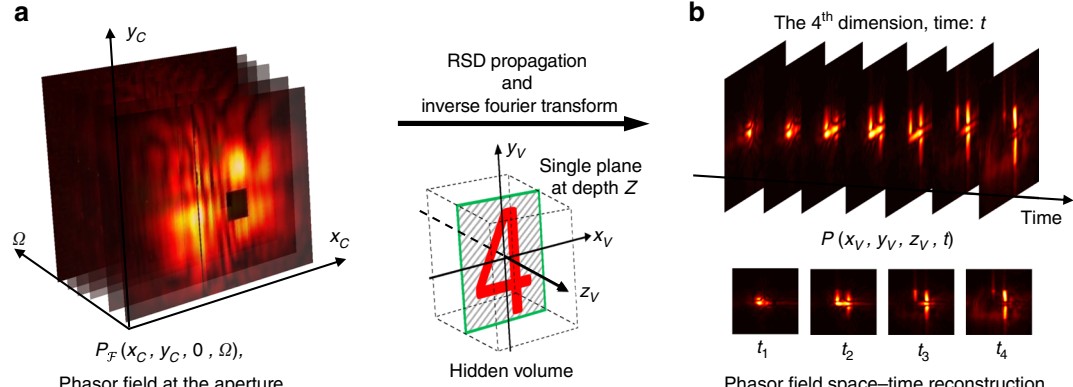

**Fig. 3 Space-time wave propagation using RSD. a** The phasor field collected at the aperture forms a spatial frequency cube. Given the output plane, by using the RSD propagation model, we can recover the hidden wavefront at any time instance. **b** This space-time wave propagation method where one can reveal a spherical wavefront that moves into the hidden scene. Even though (**b**) only shows reconstruction at a single depth plane, our proposed method can be generalized into the three-dimensional volume as well, which leads to a four-dimensional reconstruction space-time volume.

transform at the plane should be calculated with a different time shift at each voxel. In order to circumvent this tedious process, it is reasonable to split the plane into sections and within each section use the same (artificially corrected) travel time. Since the virtual illumination pulse has a finite temporal width, there can be significant overlap between a RSD reconstruction plane and the piece-wise pulse time shift. The spatial sectioning, i.e., the assignment which spatial region is reconstructed with the same time shift, is illustrated in Fig. 4.

Our objective thus is to reconstruct each voxel at depth $z$ at a time $t$ when it is actually illuminated by the virtual pulse. We first define the spatial pulse width $D = c \cdot \sigma 0.15$. In the next step, the radial difference between any voxel on the reconstruction plane and the maximum of the pulse tangential to the reconstruction is calculated by

$$E(x_v, y_v, z_v) = \sqrt{(x_v - x_{ls})^2 + (y_v - y_{ls})^2 + z_v^2} - z_v. \quad (11)$$

The geometry is shown in Fig. 4, where only the 3D cross-section at $y_v = 0$ is displayed. The spatial sectioning is defined via the functions

$$M_1(x_v, y_v, z_v) = \begin{cases} 1 & 0 \leq E(x_v, y_v, z_v) \leq \frac{D}{2} \\ 0 & \text{else} \end{cases}, \quad B_1 = \tilde{D},$$

$$M_2(x_v, y_v, z_v) = \begin{cases} 1 & \frac{D}{2} < E(x_v, y_v, z_v) \leq \frac{3D}{2} \\ 0 & \text{else} \end{cases}, \quad B_2 = \tilde{D} + D,$$

$$\vdots$$

$$M_L(x_v, y_v, z_v) = \begin{cases} 1 & (L-\frac{3}{2})D < E(x_v, y_v, z_v) \leq (L-\frac{1}{2})D \\ 0 & \text{else} \end{cases}, \quad B_L = \tilde{D} + (L-1)D,$$

$$(12)$$

which also tell us which spatial regions have to use which distance shift $B_1, \ldots, B_L$. The virtual illumination pulse illuminates a spherical shell of thickness $D$ that moves outward with time. The red and green shells in Fig. 4 illustrate the pulse positions at two different time instances, spatially separated by $D$. Depending on the distance between reconstruction voxel $(x_v, y_v, z_v)$ and the pulse maxima, this voxel will get assigned the time of the closest pulse maximum. Note that planes at a larger distance $z_v$ will have larger central regions that are treated with the same time shift. For example, a reconstruction plane far away from the relay wall may lie completely inside the shell of a single pulse and is therefore not split into sections. The described arrival time correction therefore accounts for the difference between the $z$-coordinate of a voxel and its distance to the virtual illumination source that determines the time $t$ when it is illuminated. Allowing

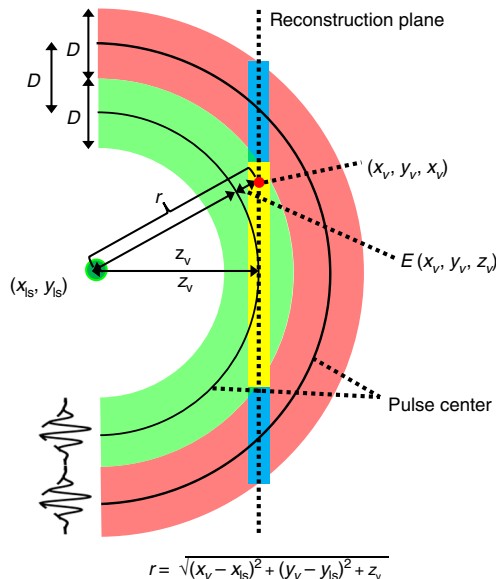

$$r = \sqrt{(x_v - x_{ls})^2 + (y_v - y_{ls})^2 + z_v}$$

**Fig. 4 Spatial sectioning for determining the piece-wise time offset.** As the yellow section of the reconstruction plane is closer to the maximum of the green pulse, it will be assigned the travel time of the green pulse. The light blue parts are closer to the maximum of the red pulse and therefore get assigned the travel time of the red pulse.

for a range $D$ around the pulse means that not the maximum of the Gaussian illumination but a point near the maximum with a somewhat lower magnitude is used. The mismatch between reconstruction planes and illumination spheres therefore only results in differences in reconstruction brightness, but not in reconstructed scene geometry.

Since $D$ is usually on the order of 20–30 cm, most simple scenes considered in this paper can be reconstructed using a single spatial section ($L = 1$). Larger field of view examples such as the Office Scene in the result section require two spatial sections ($L = 2$). The reconstruction of a larger field of view scenario using one spatial section will have a vignetting effect as if this virtual imaging system had a poor imaging quality due to a oversimplified lens design. This vignetting effect is shown in Fig. 5: On the left of the figure, one distance shift $B_1$ is used for reconstructing both spatial regions $M_1$ and $M_2$ which is equivalent to using one section. On the right, two different

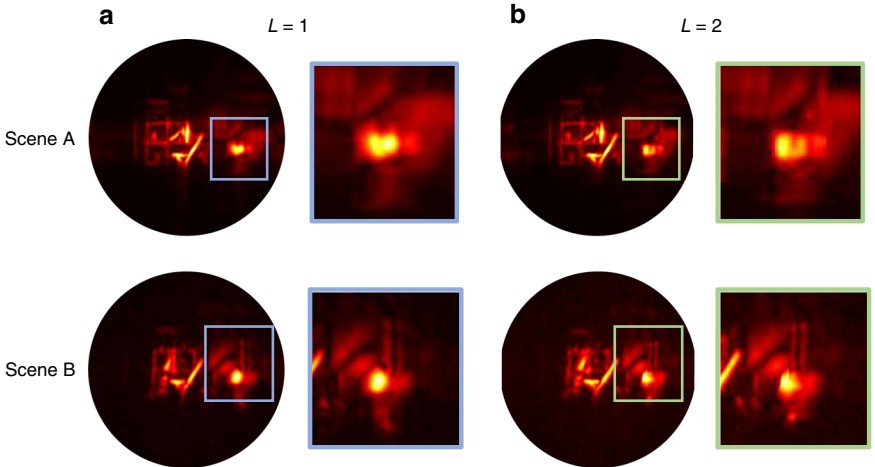

**Fig. 5 Larger field of view scenario with two versions of the office scene.** Virtual lens vignetting effect without spatial sectioning (column **a**), with spatial sectioning (column **b**). Voxels at a given depth $z$ should be computed for a time $t$ when they are actually illuminated by part of the virtual illumination pulse. Using only one distance-time shift ($L = 1$ in Eq. (12) for all voxels results in large errors in voxel brightness that lead to a blurry appearance. Using two different distance-time shifts ($L = 2$) in different voxel regions results in crisper images. This motivates the spatial sectioning method to mimic a perfect virtual imaging system that is not limited to a small field of view.

distance shift values $B_1$ and $B_2$ (see Eq. (12)) are used for $M_1$ and $M_2$.

Equation (12) contains the constant offset parameter $\tilde{D}$. This is zero for a perfectly calibrated system, such as a simulated scene, but can be adjusted to a nonzero value to account for hardware calibration in real-world experiments.

Then, the overall scene reconstruction (see Eqs. (2) and (9)) can be written as

$$
I(\mathbf{x}_v, t) = \left| \int_{\Omega_C - \Delta\Omega}^{\Omega_C + \Delta\Omega} \sum_{l=1}^{L} M_l(x_v, y_v, z_v) \cdot \exp\left( i \frac{\Omega}{c} (z_v + B_l) \right) \right.
$$
$$
\left. \cdot \left( \underbrace{\mathcal{P}_{\mathcal{F}}(x_c, y_c, 0, \Omega) * G(x_c, y_c, z_v, \Omega)}_{\text{2D convolution, implemented as 2D FFT}} \right) \frac{\mathrm{d}\Omega}{2\pi} \right|^2 .
$$
(13)

The functions $M_l(x_v, y_v, z_v)$ cut out spatial regions and tell us where to use which distance shift $B_l$, $l = 1, \ldots, L$; wherever $M_l(x_v, y_v, z_v)$ is 1, the corresponding $B_l$ is used.

Performing the NLOS reconstruction with the described RSD operator has some other advantages apart from low time and memory requirements, as shown in the results section (Section 3). The RSD calculation is easily parallelizable, because the reconstructions at different plane depths $z_v$ do not depend on each other, and Eq. (13) can be applied to each plane separately. This is in contrast to the LCT and the FK Migration methods, which perform 3D Fourier transforms of the acquired confocal data. For the RSD, when performing the reconstructions not in parallel, but starting at the relay wall and subsequently proceeding to larger depths, the memory requirement can be drastically reduced. For deriving 3D images of the reconstructed scene, it is sufficient to calculate the maximum of all reconstruction voxels along $z_v$ and store its index. This is a sparse representation of the full 3D data volume. When reconstructing by moving away from the relay wall, only the current maxima and indices of the respective $z_v$ voxel columns need to be stored, and not the full 3D reconstruction results which would require gradually increasing memory.

For all reconstructions, only a certain number of discrete frequencies $\Omega$ in the interval $[\Omega_C - \Delta\Omega, \Omega_C + \Delta\Omega]$ is propagated. It is important to point out how the number of Fourier

components that are used for reconstruction is defined. The variable $\beta$ determines the number of wavelengths $\lambda$ that fit into one pulse; $D = \beta\lambda$. The larger $\beta$, the smaller the width of the frequency domain Gaussian. $\gamma$ is the peak ratio, i.e., only the frequency components with amplitude higher than $\gamma$ are propagated. The smaller ones are neglected because they hardly contribute to the overall signal. Throughout the paper, we set $\gamma$ to 0.01, meaning that all frequency components with magnitude smaller than 1% of the maximum magnitude are ignored.

The discrete spacing $\Omega_{\text{res}}$ of the considered frequency components is given by the FFT frequency resolution:

$$
\Omega_{\text{res}} = 2\pi \frac{f_{\text{sampling}}}{N_{\text{bins}}} ,
$$
(14)

where $f_{\text{sampling}}$ is the sampling frequency of the histograms (i.e., 1/ bin width) and $N_{\text{bins}}$ the number of time bins.

The number of Fourier components depends on the choice of the virtual illumination pulse. In large scenes, it would also increase with scene depth which would increase computational and memory complexity. To avoid this, large scenes would have to be reconstructed in multiple depth sections. In this work, we reconstruct scenes with depth up to 3 m representing the largest complex scenes for which data exist. For these scenes, a depth sectioning step is not necessary.

**Fourier domain histogram (FDH) single photon capture.** According to Eq. (7), the virtual wave acquired at the virtual aperture is calculated by $\mathcal{P}_{\mathcal{F}}(\mathbf{x}_c, \Omega) = \mathcal{P}_{\mathcal{F}}(\mathbf{x}_p, \Omega) \cdot H_{\mathcal{F}}(\mathbf{x}_p \rightarrow \mathbf{x}_c, \Omega)$. This requires the Fourier domain representation of the impulse response $H_{\mathcal{F}}(\mathbf{x}_p \rightarrow \mathbf{x}_c, \Omega)$ from $\mathbf{x}_p$ to $\mathbf{x}_c$. A new memory efficient direct acquisition method for $H_{\mathcal{F}}(\mathbf{x}_p \rightarrow \mathbf{x}_c, \Omega)$ is presented in the following.

The SPAD detector uses time-correlated single photon counting (TCSPC) to generate the transient responses $H(\mathbf{x}_p \rightarrow \mathbf{x}_c, t)$. After the emission of a laser pulse, a SPAD pixel receives one photon and an electronic signal is transmitted to the TCPSC unit that encodes the time between the emission of the laser pulse and the detection of an associated returning photon. The arrival times of all photons during a measurement interval are transferred to a computer and are arranged in a histogram to obtain the transient scene response $H(\mathbf{x}_p \rightarrow \mathbf{x}_c, t)$ for a given $\mathbf{x}_p$

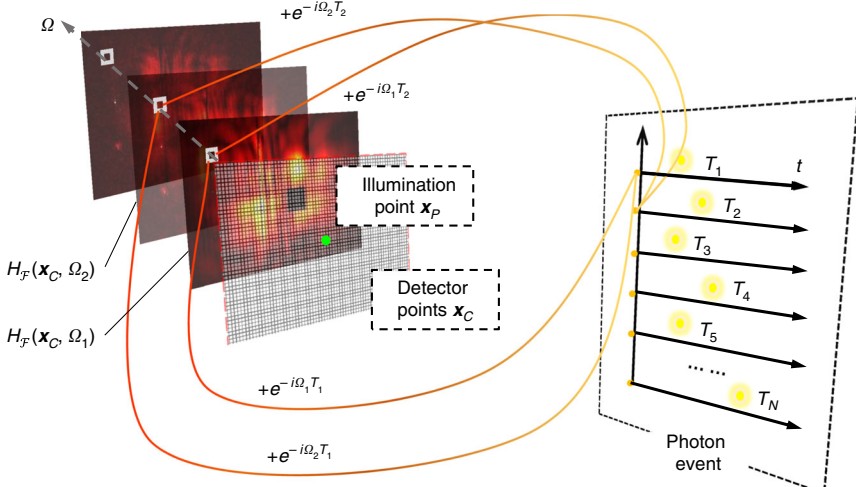

**Fig. 6 Illustration of Fourier Domain Histogram.** Instead of binning the photon event in time, we propose doing the binning in the frequency domain. This allows us directly to sample the phasor field wavefront $H_{\mathcal{F}}(\mathbf{x}_c, \Omega)$ used for reconstructions. $\Omega$ stands for the frequency range for the phasor field wavefront. The equation for the Fourier Domain Histogram can be applied during measurements, which is a summation of complex phasors (or a separated real and imaginary part).

and $\mathbf{x}_c$. To obtain $H_{\mathcal{F}}(\mathbf{x}_p \rightarrow \mathbf{x}_p, \Omega)$ we could collect and store these TCSPC histograms and perform the Fourier transform on it. A more memory efficient way is to build the frequency spectrum directly from the timing data obtained from the hardware. We call this new capturing method a FDH and its creation process is shown in Fig. 6. It can be written as

$$
\begin{aligned}
H_{\mathcal{F}}(\mathbf{x}_p \rightarrow \mathbf{x}_c, \Omega) &= \int_{-\infty}^{+\infty} H(\mathbf{x}_p \rightarrow \mathbf{x}_c, t) \cdot e^{-i\Omega t}\, \mathrm{d}t \\
&= \int_{-\infty}^{+\infty} \left( \sum_{n=1}^{N} \delta(t - T_n) \right) \cdot e^{-i\Omega t}\, \mathrm{d}t \\
&= \sum_{n=1}^{N} e^{-i\Omega T_n}.
\end{aligned}
\tag{15}
$$

The travel times $T_n$ are discrete; the time resolution is determined by the acquisition hardware (in the context of NLOS imaging typically a few to tens of picoseconds). Equation (15) means that the FDH $H_{\mathcal{F}}(\mathbf{x}_p \rightarrow \mathbf{x}_c, \Omega)$ is acquired by multiplying each of the $N$ photon travel times $T_n$, $n = 1, \ldots, N$, by a phase term depending on the considered frequency $\Omega$ and adding the result to the previous value for that frequency. As a consequence, instead of a large number of time bins (on the order of thousands), only one value for each $\Omega$ (typically dozens, as shown in the results in Section 3) needs to be stored and processed. Figure 6 illustrates the generation of the FDH. Similar to the time domain histogram binning, this FDH performs binning for each captured photon.

We want to remark that the travel times $T_n$ in Eq. (15) are measured from the respective illumination position on the relay wall into the scene and back to the relay wall at the detector focus position. The travel times from the laser setup to the illumination on the relay wall and from the detector focus point on the relay wall to the detector setup have been subtracted and are not part of $H$. Alternatively, the total travel time from laser to detector can be incorporated and the travel times from laser to wall and wall to detector are combined into $\Delta t$. The final result from Eq. (15) is then multiplied by $e^{i\Omega \Delta t}$ to correct for this constant time offset.

**Phasor field NLOS camera for confocal measurements**. The RSD reconstruction method for NLOS data presented so far only deals with the non-confocal case, which means that the illumination point $\mathbf{x}_p$ and the camera point $\mathbf{x}_c$ on the relay wall are different. However, a confocal dataset $H^c(\mathbf{x}_p \rightarrow \mathbf{x}_c, t)$ as used in LCT and FK migration algorithms[8,9] only contains data with $\mathbf{x}_p = \mathbf{x}_c$:

$$
H^c(\mathbf{x}_p \rightarrow \mathbf{x}_c, t) = H(\mathbf{x}_p \rightarrow \mathbf{x}_c, t)\delta(\mathbf{x}_p - \mathbf{x}_c). \tag{16}
$$

Such a dataset is not suitable for implementing the virtual point light source described in Eq. (5). Instead, we can model an illumination wavefront that is focused on $\mathbf{x}_v$:

$$
\mathcal{P}(\mathbf{x}_p, t) = e^{i\Omega(t - \frac{1}{c}|\mathbf{x}_v - \mathbf{x}_p|)} e^{-\frac{(t - t_0 - \frac{1}{c}|\mathbf{x}_v - \mathbf{x}_p|)^2}{2\sigma^2}}. \tag{17}
$$

Setting $t_0$ to 0 and applying the Fourier transform leads to

$$
\mathcal{P}_{\mathcal{F}}(\mathbf{x}_p, \Omega) = \left( 2\pi \delta(\Omega - \Omega_C) \underset{f}{*} \sqrt{2\pi}\sigma e^{-\frac{\sigma^2 \Omega^2}{2}} \right) e^{-i\frac{\Omega}{c}|\mathbf{x}_v - \mathbf{x}_p|}. \tag{18}
$$

Inserting into Eq. (7) yields

$$
\begin{aligned}
I(\mathbf{x}_v, t) &= \\
&\left| \int_{-\infty}^{+\infty} e^{i\Omega t} \mathcal{R}_{\mathbf{x}_v} \left( \int_P (2\pi)^{\frac{3}{2}} \sigma e^{-\frac{\sigma^2(\Omega - \Omega_C)^2}{2}} e^{-i\frac{\Omega}{c}|\mathbf{x}_v - \mathbf{x}_p|} \delta(\mathbf{x}_p - \mathbf{x}_c) \cdot H_{\mathcal{F}}(\mathbf{x}_p \rightarrow \mathbf{x}_c, \Omega) \mathrm{d}\mathbf{x}_p \right) \frac{\mathrm{d}\Omega}{2\pi} \right|^2 \\
&= \left| \int_{-\infty}^{+\infty} e^{i\Omega t} \mathcal{R}_{\mathbf{x}_v} \left( (2\pi)^{\frac{3}{2}} \sigma e^{-\frac{\sigma^2(\Omega - \Omega_C)^2}{2}} e^{-i\frac{\Omega}{c}|\mathbf{x}_v - \mathbf{x}_c|} \cdot H_{\mathcal{F}}(\mathbf{x}_c \rightarrow \mathbf{x}_c, \Omega) \right) \frac{\mathrm{d}\Omega}{2\pi} \right|^2 \\
&= \left| \int_{-\infty}^{+\infty} e^{i\Omega t} \int_C (2\pi)^{\frac{3}{2}} \sigma e^{-\frac{\sigma^2(\Omega - \Omega_C)^2}{2}} e^{-i\frac{\Omega}{c}|\mathbf{x}_v - \mathbf{x}_c|} \cdot H_{\mathcal{F}}(\mathbf{x}_c \rightarrow \mathbf{x}_c, \Omega) e^{-ik|\mathbf{x}_v - \mathbf{x}_c|} \mathrm{d}\mathbf{x}_c \frac{\mathrm{d}\Omega}{2\pi} \right|^2 \\
&= \left| \int_{-\infty}^{+\infty} e^{i\Omega t} \int_C (2\pi)^{\frac{3}{2}} \sigma e^{-\frac{\sigma^2(\Omega - \Omega_C)^2}{2}} \cdot H_{\mathcal{F}}(\mathbf{x}_c \rightarrow \mathbf{x}_c, \Omega) e^{-2ik|\mathbf{x}_v - \mathbf{x}_c|} \mathrm{d}\mathbf{x}_c \frac{\mathrm{d}\Omega}{2\pi} \right|^2.
\end{aligned}
\tag{19}
$$

The reconstruction thus uses an RSD operator with an additional factor of two doubling all distances. We use our fast RSD operator to evaluate this RSD integral.

Both the computational implementation steps and the pseudocode of the presented RSD NLOS reconstruction algorithm are available in Supplementary Note 3.

**Acquisition hardware**. Most of our results in Figs. 7 and 8 are obtained on a publically available dataset[10]. In addition we provide three additional datasets (Fig. 8 rows 1, 2, and 4). The experimental setup used to create all those datasets consists of a gated single-photon avalanche diode (SPAD) with a Time-

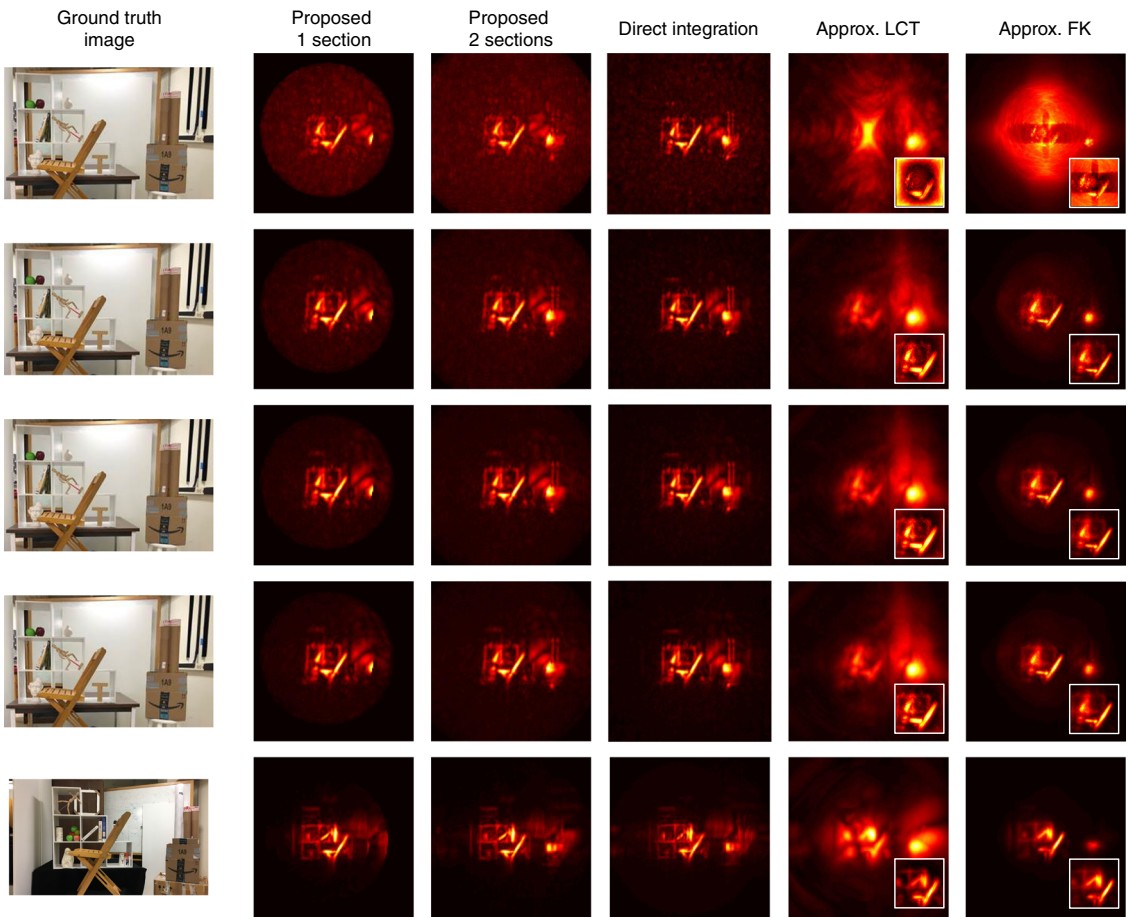

**Fig. 7 Methods comparison on Office Scene: Exposure time per each pixel measurement from first row to last row is 1 ms, 5 ms, 10 ms, 20 ms, 1000 ms (note that the 1000 ms Office Scene dataset was acquired with slight differences in the object location).** The total acquisition time from first row to last row is 23 s, 117 s, 4 min, 8 min, 390 min. The width of the reconstruction cube size in each dimension is 3 m as given along with other reconstruction parameters in Supplementary Table 4. Each column shows the reconstruction with different methods. The first two columns stand for our proposed RSD based solver with one or two spatial sections. The circle in the first column is actually the size of the farthest reconstruction plane which is the one with the largest region that is calculated with the same distance shift $B_1$. All planes in front of this one have a smaller reconstruction area; due to the maximum operation along the depth dimension, the circle size is defined by the largest one in the back. The third column is the Direct Integration (back-projection solver) as a comparison for the first two columns. The last two columns refer to the approximation method[9] which approximate non-confocal by confocal data and solve it through the scanning-based solver (LCT: forth column, FK-migration: fifth column). For the last two columns, each small image shows the results from midpoint approximation[9] in order to approximate confocal data from non-confocal measurements. The respective larger image results from zero-padding applied to the input data to show the same reconstruction volume as the first three columns.

Correlated Single Photon Counter (TCSPC, PicoQuant Hydra-Harp) with a time resolution of about 30 ps and a dead time of 100 ns to measure the time response as well as a pico-second laser (Onefive Katana HP amplified diode laser with 1 W at 532 nm, and a pulse width of about 35 ps used at a repetition rate of 10 MHz) as light source. The entire system's temporal resolution is around 70 ps. We perform several new non-confocal experiments (20 ms exposure time Office Scene in Supplementary Table 1 and two scenes showing patches 4 and 44i, Supplementary Table 2) and use existing experimental non-confocal[10] and confocal[9] datasets to compare our method against the literature. The experiments are performed using the non-confocal acquisition scheme. The detection aperture on the relay wall is around 1.8 m by 1.3 m with 1 cm spacing between each captured time response. This yields 181 by 131 captured time responses for each scene. Scene descriptions are provided in Supplementary Table 3 including scene depth complexity and target materials.

**Reconstructions.** Reconstructions with maximum intensity projection along the depth direction are shown in Figs. 7 and 8.

Results with three-dimensional volume rendering are shown in Supplementary Fig. 2. For the non-confocal dataset, we consider three solvers: our proposed fast RSD based solver, the back-projection solver presented previously[10] (denoted by Direct Integration) and two approximate fast methods: LCT and FK Migration, referred to as approx LCT and approx FK. Both of them cannot operate on the non-confocal data used here, however, Lindell et al.[9] describe a way to turn a non-confocal dataset into an approximately confocal dataset that allows application of non-confocal methods. We implemented this approximate method based on the description and show the approximate reconstructions in the last two columns of Figs. 7 and 8. We refer to this approximation as midpoint approximation. The Direct Integration solver is slow because of the discrete integration step, but we use it as an accurate theoretical calculation reference for our method. Both approx LCT and approx FK yield blurry results compared to both the proposed RSD and Direct Integration when applied to the single plane targets shown in Fig. 8. Beside the simple plane scenes, we consider a more complex Office Scene with multiple targets and targets outside the scanning aperture

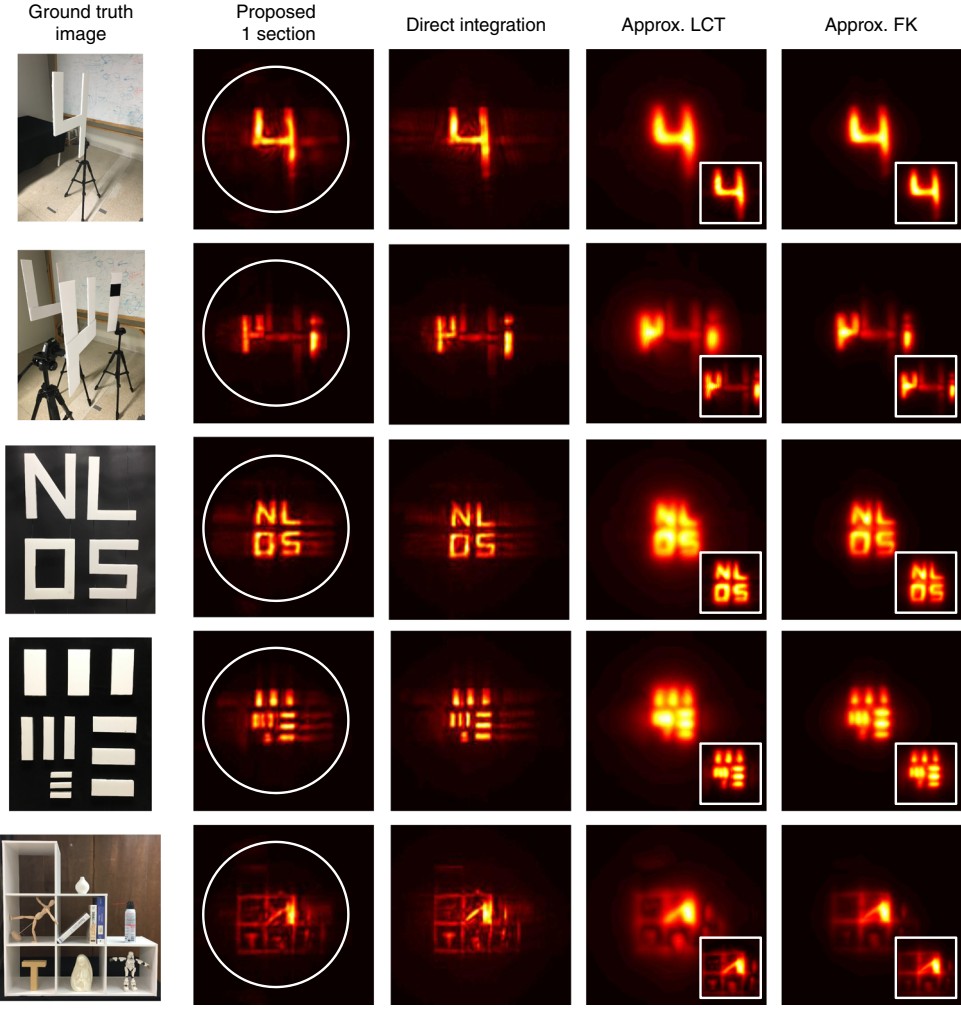

**Fig. 8 Methods comparison on simple targets: Exposure time for these scenes are all 1000 ms per each pixel measurement.** The total acquisition time for each target is 390 min. The width of the reconstruction cube size in each dimension is about 2 m as given along with other reconstruction parameters in Supplementary Table 4. Each row shows a different simple target, each column the reconstruction from different methods. The first column stands for our proposed RSD based solver with one spatial section (inside white circle) corresponding to Eq. (13). The second column is from Direct Integration (back-projection solver) for comparison with the first column. The last two columns show the approximate method[9] which approximate non-confocal as a confocal datasets and reconstruct through confocal solvers (LCT: forth column, FK-migration: fifth column). For the last two columns, each small image shows the results from midpoint approximation[9] in order to approximate confocal data from non-confocal measurements. The respective larger image results from zero-padding applied to the input data to show the same reconstruction volume as the first three columns.

with a large field of view. The results are shown in Fig. 7. None of the approximate solutions achieves the imaging quality of the phasor field solution (first three columns in Fig. 7). There are two properties of the approximate solutions: The LCT and FK Migration methods inherently can only recover objects within the aperture, and, to make things worse, the approximation made by converting non-confocal into confocal datasets results in an even smaller aperture. To recover a larger hidden volume with a larger field-of-view of the virtual image, we perform a zero padding step at the aperture to make it larger. Even in this case, none of the approximate solutions provides sharper and well-focused images than the RSD-based reconstruction algorithms.

One thing we would like to point out about Fig. 7 for our proposed method is that as the exposure decreases down to 1 ms, the calculation error is highlighted as a almost constant background. We can reduce this artifact in the short exposure scenario by increasing the number of used Fourier components to mimic a shorter Gaussian envelope for the illumination pulse. This effect of choosing a different number of Fourier components

for the final results as well as the corresponding execution time is also shown in Fig. 9 on the 20 ms Office Scene.

For both simple scene and complex Office Scene results, our proposed methods are much faster with reconstructions in seconds. The exact run times of the un-optimized solvers discussed above are given in Supplementary Tables 1 and 2. All computational parameters (number of Fourier components for the new RSD method, reconstruction volume size, voxel grid resolution etc.) used for creating the reconstruction results in Figs. 7 and 8 are provided in Supplementary Table 4.

## Discussion

To the best of our knowledge, our proposed method is the first to solve the general non-confocal NLOS imaging scenario with a similar time requirement and computational complexity as the fastest existing algorithms. In contrast to them, however, our method has much lower memory requirements. This allows us to reconstruct larger scenes and will enable implementation on

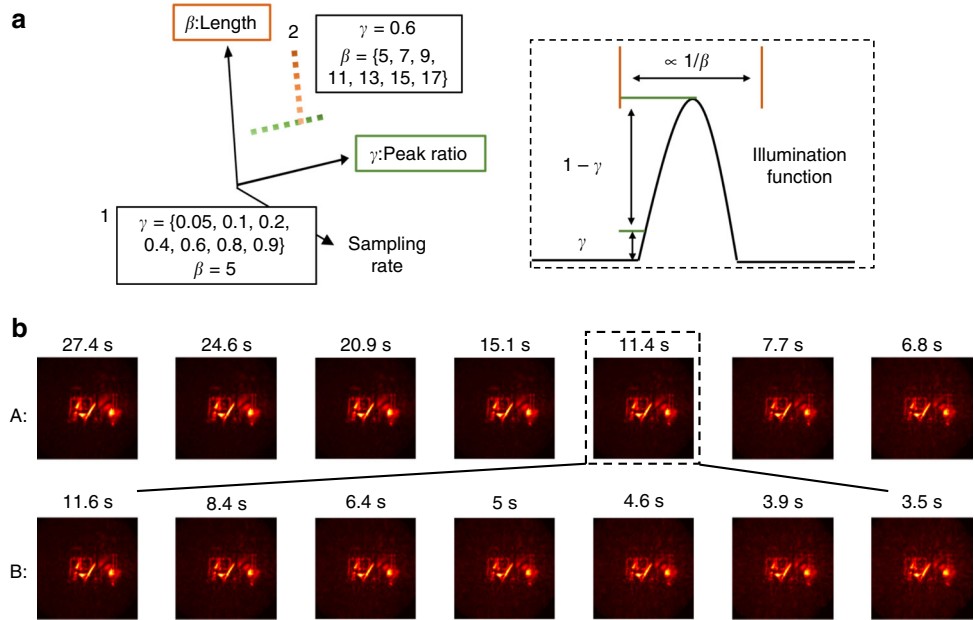

**Fig. 9 Virtual illumination function design space and reconstruction speed. a** The virtual illumination function design space and **b** the corresponding design parameters with reconstructions and their run times. As shown in **a**, considering the peak ratio $\gamma$ and the temporal envelope length $D = \beta \cdot \lambda$ (characterized by how many modulation cycles fit inside the envelope) for the illumination function, we plot each corresponding reconstruction and processing time for the Office Scene dataset with 20 ms exposure in **b**. The peak ratio $\gamma$ coefficient is used for thresholding in the Fourier domain. Overall, the more Fourier components used during the reconstruction, the better the noise reduction and the longer the calculation time.

embedded systems and GPU units where memory is limited. We believe our method will enable real time NLOS imaging and reconstruction of large room scale scenes at full resolution. In this section, we discuss some related NLOS imaging works which currently fail to support real time NLOS and the computational complexity of our proposed method.

We discussed some related works which currently cannot support real time NLOS imaging scenarios. Such reconstruction methods include a fast GPU backprojection solver[18]. This method solves the back-projection method faster than CPU implementations, but is still too slow to operate in real time, partially to high memory bandwidth requirements. The current implementation also does not support negative numbers and double precision, both of which are necessary for more advanced phasor field backprojection applications. First returning photon and Fermat path theory can recover surface geometry[19,20] of simple scenes with single objects. Improved iterative back-projection solutions using a new rendering model and frequency analysis[21,22] can create particularly high quality surface reconstructions. Full color NLOS imaging with single pixel photomultiplier tube combined with a mask[23,24] has also been demonstrated. Further work includes real-time transient imaging for amplitude modulated continuous wave lidar applications[25], analysis of missing features based on time-resolved NLOS measurements[26], convolutional approximations to incorporate priors into FBP[27], occlusion-aided NLOS imaging using SPADs[28,29], Bayesian statistics reconstruction to account for random errors[30], temporal focusing for a hidden volume of interest by altering the time delay profile of the hardware illumination[31], and a database for NLOS imaging problems with different acquisition schemes[32]. Reconstruction times for all these methods remain in the minutes to hours range even for small scenes of less than a meter in diameter. To the best of our knowledge, none of the works above have been applied successfully to larger and more complex scenes with the exception of the back-projection based methods. Ahn et al.[27] can improve the reconstruction quality after the back-projection via an iterative convolution step. Since the method involved a back-projection as it's first step it shares the speed and complexity disadvantages of the back-projection based methods mentioned above. In addition, the resolution of an NLOS reconstruction is limited by the time resolution of the detection system[8]. For a SPAD, the time resolution is 30 ps at best leading to a theoretically achievable grid resolution of 1 cm in the hidden scene. Methods that can process scenes of moderate and high volume and complexity include FK Migration, the LCT, and Phasor-Field virtual waves which are discussed in this paper.

There are also several contributions showing that it is possible to do NLOS imaging without picosecond scale time resolution or with non-optical signals: Inexpensive nanosecond time of flight sensors can be used to recover the hidden scene[33], tracking can be performed using intensity based NLOS imaging [34], occlusions are harnessed to recover images around a corner using regular cameras[35–37], even describing the occlusion-aided method as a blind deconvolution problem without knowledge of the occluder[38]. Other approaches decode the hidden object from regular camera images by using a deep neural network trained with simulated data only[39], or use acoustic[40] or long-wave infrared[41] signals to image around the corner. While promising for low cost applications, none of these methods achieve reconstruction qualities comparable to the picosecond time-resolved NLOS imaging approaches.

Our proposed method is computationally bounded by the FFT process. Let $N$ denote the number of pixels along each of the three spatial dimensions of the reconstruction space. Calculating the RSD reconstruction requires a 2D FFT at each of the $N$ depth planes for each Fourier component. The computational complexity of the presented algorithm is then given by

$$\mathcal{O}(N^3 \log N) \qquad (20)$$

because the number of Fourier components is just a constant by performing reconstructions in multiple depth sections which is shown in Section "Fast Phasor Field Diffraction". LCT and FK have the same complexity as described in the respective papers[8,9]; all other methods applied to complex scenes published so far have

higher complexity. The memory complexity of our algorithm is defined by the need to store the FDH (details provided later in the Methods section) and the resulting 2D image. For the scenes described here this is actually $O(N^2)$. In larger scenes we would need to store multiple FDHs for multiple depth sections in order to maintain low computational complexity. In this case memory complexity is $O(N^3)$.

The computational complexity for our proposed solution, LCT and FK are the same from the theoretical point of view. In practice, due to the need for oversampling and interpolation the actual memory requirement for each method is several hundred times higher than ours in their current form. Unfortunately there are many different options with different trade-offs and it is not completely clear which is used inside the Matlab interpolation functions used in the current algorithms. Existing papers on FK Migration typically discuss their particular choices and their impact on memory complexity and reconstruction quality in considerable detail[42,43]. To get a better understanding of the source of the memory requirements, let us consider the requirements for our method and FK Migration as an example.

Consider a scene with the size of the Office Scene from 0 to 2.5 m away from the relay wall that is used in this paper. The temporal measurements are collected from 150 by 150 spatial sampling points. For our proposed method, storing the FDH requires 150*150*139*4 (139 is the number of used frequency components for the similarly sized Office Scene as shown in Supplementary Table 4) bytes which is around 12.51 MB (or 25 MB to store both real and imaginary parts). Algorithms exist that can compute the FFT without requiring extra working memory. Our reconstructions are computed slice by slice and only the maximum is kept. The only additional memory required is to store the 2D result. If we would like to create a 3D visualization, we have to store the index of the maximum. This requires 150*150*4*2 = 180 kB of additional memory. The total memory required is thus 50.18 MB.

FK Migration needs the histogram in the time domain. Sampling resolution in the histogram and resolution in the Fourier domain are linked through the FFT and cannot be chosen freely. To cover this scene setup with 32 ps temporal sampling rate, at least 512 temporal sampling bins are required for each captured time response to cover the light path round trip in 5 m. Assume each temporal bin is in single-precision using 4 bytes. For LCT/ FK, one needs to store 150*150*512*4 bytes which is around 46 MB. This is already significantly more than the memory requirement of our entire algorithm. We assume the 3D FFT can be performed without additional working memory. For FK and LCT, two extra steps are required apart from the 3D FFT. The first extra step is to oversample the DFT by zero padding the data before the Fourier transform. This provides higher resolution in the Fourier domain and makes the following interpolation step easier. The current implementation increases the size of the data by 2 in each dimension by zero padding resulting in a memory need of 0.368 GB ($2^3$*46 MB). This 3D dataset structure is complex-valued and needs an additional second channel to store real and imaginary part. The second extra step is to perform the interpolation to compute the points in Fourier space that are needed as input for the inverse 3D FFT. As is stated directly in the literature, this interpolation step is the complexity bottleneck for FK Migration[42]. The current FK Migration code uses the Matlab function interp[9]. That uses two neighbor points along each dimension to perform a linear 3D interpolation. Without prior assumptions about the structure of the grids, search in nearest neighbors would have a computational complexity for $O(N^6)$ which is impractical. This can be improved by pre-computing a map of nearest neighbors using a faster algorithm like a k-d tree.

To store the six nearest neighbors of each data point requires 2.21 GB (6*$2^3$*46 MB). Then the linear interpolation if implemented in this way would require 2.21 GB of working memory in addition to the size of the data itself. While we can't be sure that this is what Matlab is doing, the memory load is consistent with our measurements. The memory profile while running both methods on our captured dataset is shown in Supplementary Fig. 5 and its order of magnitude coincides with the estimate. After inverse 3D Fourier transform the final result is a sparse three-dimensional complex matrix of size 150*150*300 or larger. The current method reconstructs a higher resolution matrix as a side effect of the oversampling. This is not actually needed and doesn't significantly affect the result. We thus have an additional memory need of 150*150*300*2*4 = 54 MB. Note again that just the result takes up more memory than our entire computation. This results in a total peak memory use of 2.21 GB + 2*0.368 GB = 2.946 GB. There are several ways that can likely reduce this memory load.

Knowledge of the relative layouts of the two grids may reduce or eliminate the requirement for working memory in the interpolation. One can also fine tune the trade-off between Fourier domain oversampling, more sophisticated interpolation methods, and reconstruction quality. It might also be possible to perform further down-sampling along the temporal dimension and use single instead of double precision variables to require less memory. These approaches are interesting topics for future research and can draw from considerable prior work on this problem in related FK Migration application areas. At present, however, the method takes several hundred times more memory than our proposed method. The LCT includes a similar re-sampling step that creates large memory requirements. Re-sampling and interpolation problems in this domain are studied in the literature covering planar and spherical inverse radon transforms.

## Methods

**Discrete phasor field diffraction model and implementation**. In this section, the computational implementation for the model derived in the main paper is described. We will explain the discrete RSD model and implementation here; the respective pseudocode is provided in Supplementary Note 3. We introduce the RSD discrete model and link it to physical measurement parameters (scanning aperture size, sensor grid spacing) and then provide the corresponding algorithmic implementation procedure as a guideline.

We provide a description for the FFT based RSD solver implementation for $\mathcal{R}_{z_v}(\cdot)$ in Eq. (8). For an actual algorithm implementation, it is necessary to discretize the continuous model. Considering discrete parameters such as a finite size square aperture sampling both the camera aperture and reconstruction planes $C$ and $V$ at uniform distances $\delta_{in}$ and $\delta_{out}$, the wavefront is a matrix of size $N \times N$. We use the symbols $[nx_v, ny_v]$ and $[nx_c, ny_c]$ to represent the discrete indices. We consider $\delta_{in} = \delta_{out} = \delta$ spatial sampling in both input and output domains where $\delta = \frac{\lambda}{2}$ is the maximum sampling distance[44]. The variable $Z$ denotes the maximum value of $z_v$. For brevity, all following equations ignore the frequency variable $\Omega$ of the input $\mathcal{P}_{\mathcal{F}}[nx_c, ny_c]$ and output $\mathcal{P}_{\mathcal{F}}[nx_v, ny_v]$ wavefronts. Overall, with these discrete parameters, the RSD operator in Eq. (8) can be written as a standard discrete convolution as follows:

$$\mathcal{P}_{\mathcal{F}}[nx_v, ny_v] = \sum_{nx_c, ny_c = -N/2}^{N/2-1} \sum^{N/2-1} \mathcal{P}_{\mathcal{F}}[nx_c, ny_c] \cdot G[nx_v - nx_c, ny_v - ny_c, z_v]$$

$$G[nx_c, ny_c] = \alpha \cdot \delta^2 \cdot \frac{\exp[-i\frac{\Omega}{c}\sqrt{nx_c^2\delta^2 + ny_c^2\delta^2 + z_v^2}]}{\sqrt{nx_c^2\delta^2 + ny_c^2\delta^2 + z_v^2}}. \tag{21}$$

Thus, the discrete model in Eq. (21) can be implemented as two-dimensional Fast Fourier Transform (2D FFT) algorithm. Notice that the parameter $\alpha(x_v, y_v, z_v)$ is ignored for the reconstruction in Eq. (21). Then the algorithmic procedure is:

*Goal*: Given input wavefront $\mathcal{P}_{\mathcal{F}}[nx_c, ny_c]$, spacing between the input and output parallel plane $z_v$ (depth), angular frequency $\Omega$ and associated wavelength $\lambda$, calculate the output wavefront $\mathcal{P}_{\mathcal{F}}[nx_v, ny_v]$ by

$$\mathcal{P}_{\mathcal{F}}[nx_v, ny_v, \hat{z}] = \mathcal{R}_{z_v}\left[\mathcal{P}_{\mathcal{F}}[nx_c, ny_c, 0]\right]. \tag{22}$$

*Step 1*: Discretize depth

$$\hat{z} = \frac{Z}{\delta}$$

*Step 2*: Zero padding according to the desired reconstruction volume size

$$N' = \text{Hidden volume side length}$$
$$\text{pad} = \frac{N'-N}{2}$$
$$\mathcal{P}_{\mathcal{F}}[nx_c, ny_c] = \text{padarray}\left(\mathcal{P}_{\mathcal{F}}[nx_c, ny_c], [\text{pad}, \text{pad}], 0\right)$$
$$\text{Update discrete size}: \ N = N'$$

*Step 3*: Variable substitution

$$\eta^2 = \frac{\lambda Z}{N\delta^2} = \frac{\lambda \hat{z}}{N\delta}$$

*Step 4*: Compute convolution kernel

$$G[nx_c, ny_c, \hat{z}] = \frac{\exp\left[-i2\pi \cdot \hat{z}^2/(\eta^2 N) \cdot r\right]}{r}$$
$$r = \sqrt{nx_c^2/\hat{z}^2 + ny_c^2/\hat{z}^2 + 1}$$

*Step 5*: Perform the inverse diffraction

$$\mathcal{P}_{\mathcal{F}}[nx_v, ny_v, \hat{z}] = \mathbf{IFFT}\left\{\mathbf{FFT}\left\{\mathcal{P}_{\mathcal{F}}[nx_c, ny_c, 0]\right\} \bullet \mathbf{FFT}\left\{G[nx_c, ny_c, \hat{z}]\right\}\right\}$$

Here are some short explanations for the computational algorithm above:

1. In *Step 2*, to reconstruct a volume with maximum dimensions $x_v$ and $y_v$ larger than the maximum aperture dimensions $x_c$ and $y_c$ (or $x_p$ and $y_p$), one needs to increase the spatial dimension (parameter $N'$) by zero padding the input wavefront.
2. *Step 5* is based on the standard **FFT** and **IFFT** algorithm. The symbol • stands for the point-wise multiplication operation. *Step 5* can be done in space as well. However, for two matrices of comparable size, Fourier domain multiplication usually runs faster than spatial convolution.
3. The inverse focusing step realized by the RSD creates a virtual image on the other side of the relay wall, so the sign of the depth parameter $z_v$ should be chosen negative for the considered reconstruction volume.

**Memory usage**. We are also interested in the memory usage of the fast algorithms (proposed, approx LCT, approx FK). We acquire the memory usage during reconstructions for the Office Scene in Fig. 7. The memory profile during execution is shown in Supplementary Fig. 5. Our memory testing as well as all our code are running on an *Intel Core i7-7700 CPU, 3.6 GHz x 8 with 32 GB memory* using Matlab. During testing, the base memory usage for non-GUI Matlab is around 750 MB. Independent of the reconstruction quality, approx LCT and approx FK need much more memory than our proposed method. Neglecting the memory of the operating system etc., our method would require about 5 MB of memory when implemented most efficiently. A more detailed discussion regarding to the memory usage can be found later in the Discussion section.

**Confocal and rendered data**. As a confocal scanning scenario, we use the open source experimental dataset[9]. Our proposed reconstruction method requires similar time and lower memory usage compared to the LCT[8] and FK[9] methods. The reconstruction results of confocal datasets are shown in Supplementary Figs. 3 and 4. In terms of the difference between non-confocal (SPAD array) and confocal (Single SPAD with scanning) capture, we provide a short discussion in Supplementary Note 4.

Reconstructions using a rendered dataset with known ground truth are shown in Supplementary Fig. 1. Our proposed method reconstructs an image of the hidden scene that resembles the image that would be captured with a camera located at the relay wall. In our reconstructions, we recover phasor field irradiance for the hidden object. It is expected that the reconstruction shows spatial distortions similar to the ones seen by a real camera, as it is shown in Supplementary Fig. 1. If an exact depth measurement is desired, these biases would have to be calibrated. This is an interesting subject for future work.

## Data availability
The data supporting the findings of this study are available (downloaded) at: figshare repository https://doi.org/10.6084/m9.figshare.8084987, Computational Optics Group https://biostat.wisc.edu/~compoptics/phasornlos20/fastnlos.html and Standard Computational Imaging Lab http://www.computationalimaging.org/publications/nlos-fk/. The source data underlying Supplementary Fig. 5 are provided as a Source Data file.

## Code availability
The code used in this work is included in this published article and its supplementary information files.

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

## Acknowledgements

This work was funded by DARPA through the DARPA REVEAL project (HR0011-16-C-0025), and the DURIP program (FA9550-18-1-0409). We thank Jeffrey H. Shapiro for the insights about the phasor field broad-band model. We also appreciate the help of Marco La Manna, Ji-Hyun Nam, Toan Le, and Atul Ingle on the hardware setup and helpful discussion during calibrations. Xiaochun Liu would like to acknowledge the helpful discussion with David B. Lindell about his approximate non-confocal method, with Ibón Guillén and Miguel J. Galindo about volume rendering methods and simulated datasets.

## Author contributions

X.L. and A.V. conceived the method. X.L. implemented the computational model for the reconstruction. X.L., S.B. performed experiments and comparison for the paper. A.V. supervised all aspects of the project. All authors contributed to designing the experiments and writing the paper.

## Competing interests

The authors declare no competing interests.
