## [Peer Review File · Nature Communications]

Reviewers' comments:

Reviewer #1 (Remarks to the Author):

The recent Nature paper on phasor fields was a valuable contribution and very interesting. Phasor fields turn time-resolved NLOS measurements into a virtual wave and then propagate that wave to the hidden scene using a free-space propagation operator, such as the Rayleigh-Sommerfeld Diffraction Integral (Eqs. 1-4). These concepts were thoroughly explored in the recent Nature paper. Unfortunately, this manuscript adds little new. To be precise, the primary contribution of this manuscript is to implement the free-space propagation operator in the Fourier domain (Eq. 8). This is common practice in computational optics and outlined in standard textbooks, such as Goodman's Fourier Optics book, and also taught in introductory Fourier optics classes.

What is more concerning is that the authors make contradictory and false statements throughout the manuscript, possibly trying to mislead the reader and misrepresent their contributions; they also do not actually demonstrate many of their claims or quantitatively evaluate them. Therefore, I cannot support this submission.

One of the primary arguments that is made throughout the manuscript is that the proposed method has a lot lower memory requirements than other real-time NLOS algorithms. This is false, as the required 3D Fast Fourier Transforms in those methods can be sequentially applied to all three dimensions x, y, t to reduce their memory footprint by three orders of magnitude at the cost of increased compute times albeit with the same computational complexity of $O(N^3 \log N)$.

The proposed method is an approximation: Fourier-based free-space propagation followed by a simple max filter. Other methods, such as LCT and FK, are exact solutions to the inverse problem. Therefore, I don't see why the proposed method could be better for confocally scanned data (some of the results seem questionable, see comments below).

A discussion of available photon counts is missing. It is argued that real time capture is important and that SPAD arrays could achieve that. Yet it is neither shown nor likely as these arrays have significantly worse fill factors and photon detection probabilities. The photon counts of the proposed non-confocal system should actually be similar or worse than confocally scanned data.

Finally, there seems to be some pre-processing required for Phasor Fields and this should be included in the compute time calculations / performance measurements.

Examples of claims that are not demonstrated:

- The abstract motivates real-time capture with SPAD arrays, yet there is no discussion of actually capturing data with a SPAD array. Was any of the data captured in real time? Was a SPAD array used? It seems unlikely, because available SPAD arrays neither have the time resolution nor the photon detection probability (i.e., "sensitivity") to actually capture a sufficient number of third-bounce photons of diffuse room-sized scenes as claimed in the paper. No discussion of SPAD array hardware or optical setup used to capture any of the presented data is included and all of the data shown is captured, in previous work, with a single scanned SPAD.
- "SPADs can potentially be manufactured at low cost and in large arrays enabling fast parallel NLOS capture." - this statement (among others) indicates that SPAD arrays and thus real-time capture was not actually implemented, but that seemed like a major claim in the abstract and early parts of the introduction
- there is no quantitative evaluation of any of the results; it is claimed that the proposed method is

"better" and "less blurry" than other methods, but that is not quantified anywhere

Examples of false or misleading statement:

- "current reconstruction algorithms have computational and memory requirements that prevent real-time application on a desktop or embedded computer" (abstract) - as discussed below, LCT and FK demonstrated NLOS imaging in real time already
- "The fastest existing algorithms also require a point-scanning or confocal acquisition of the data making capture hardware inherently noisy, slow, and complex." - single SPADs have significantly better noise performance and fill factors than SPAD arrays; the complexity of scanning confocal systems is the same as commercial LiDAR systems
- "Existing real-time demonstrations, therefore, use retroreflecting targets and reconstruct at resolutions far below the hardware limits." - this is false as O'Toole et al. 2018 demonstrated that their LCT algorithm achieves exactly the theoretical resolution limit
- "Running on a desktop computer, our method recovers a room-sized volume in seconds and has a computational complexity equal to the fastest existing algorithms and less memory usage." - this is not new and has been demonstrated previously
- "An algorithm suitable for fast NLOS imaging must fulfill three separate requirements: The ability to use data that can be captured in real time, a computational complexity allowing for execution in a fraction of a second..." - the proposed work does not show real-time capture and only relies on non-real-time data captured in previous work; real-time algorithms have been demonstrated in the past
- "Real time reconstruction of low resolution retro-reflective scenes has also been demonstrated in a confocal scanning scenario with both LCT and FK Migration methods. However, reconstruction of higher resolution scenes with diffuse surfaces is hindered by the large memory requirements and the slow confocal capture process requiring sequential point scanning capture with a single SPAD pixel." - these previous methods require a 3D FFT; this can simply be implemented by successively Fourier-transforming each of the dimensions x, y, t to reduce the memory footprint by three-orders of magnitude at the cost of slightly increased computing times (although the same order of compute time $O(N^3 \log N)$)
- "This new method performs at speed similar to LCT and FK Migration when used on confocal data, while requiring significantly less memory" - false, see comment above; it is also not clear why non-confocal measurements would be better than confocal measurements
- The results in Figs. 11,12 are questionable. The reconstruction quality of LCT and FK seem a lot worse than what was presented for that same data in the FK paper.

Other issues:

- non-confocal measurements are being discussed as being significantly better than confocal measurements, but it seems largely unclear why that is, because the scanning and system calibration complexity seems a lot higher for non-confocal measurements.
- Fig. 5 is somewhat cryptic, it's not clear what is going on

Missing references:

- Klein et al. "Tracking objects outside the line of sight using 2D intensity images", Scientific Reports 2016 (for intensity based NLOS approaches)
- Peters et al. "Solving trigonometric moment problems for fast transient imaging", SIGGRAPH Asia 2015 (for real-time transient imaging approaches)
- Heide et al. "Non-line-of-sight imaging with partial occluders and surface normals", TOG 2019 (for occlusion-aided NLOS imaging)
- Wu et al. "Frequency analysis of transient light transport with applications in bare sensor imaging", ECCV 2012 (for iterative NLOS imaging)

Reviewer #2 (Remarks to the Author):

This submission presents an exciting and timely contribution to the field of non-line-of-sight image reconstruction algorithms. Because of its convincing results, accessible technical description, and the promised dataset/source code, this work will have a high-impact and inspire other work with eventual applications across domains.

The authors present a memory and compute-efficient approach based on Rayleigh-Sommerfeld Diffraction for non-confocal non-line-of-sight measurements. While the best results of previous confocal approaches are obtained by arguably "cheating" with retroreflective materials [27], that is by engineering the BRDF of the unknown hidden scene to substantially increase SNR of the confocal measurement setup, this work considers generic scenes with a non-confocal measurement arrangement. The presented reconstruction algorithm is well-motivated and explained in detail, even providing pseudo-code of the method itself. The authors could only have provided the measurement and the algorithm source code to go beyond what is provided at this stage. The method is well evaluated against recent reconstruction methods, with only a few open points (see these listed below), and provides both a solid theoretical framework for further algorithm development in this space, as well as, a solid baseline for additional non-line-of-sight methods.

This work should be published in a timely manner, especially to keep pace with this fast-moving field. I would like the authors to consider the following requests in the final manuscript:

1) Non-confocal acquisition: The differences between the non-confocal and confocal measurements should be highlighted early in the manuscript. I would suggest a separate paragraph/section for this comparison. Confocal measurements discard a substantial amount of measurements and this should be explicitly highlighted. I also don't quite buy the claim of prior work [27,28] that confocal detection is substantially easier to implement as gating is absolutely necessary to not suffer from massive pileup due to the direct reflection. It would significantly add to the paper if the authors describe and analyze the additional transient light transport information briefly and make compare their setup briefly to these prior works.

2) Memory requirements and runtime: The claims regarding memory and runtime are validated on a broad set of reconstructions. The only open item here is the comparison against matrix-free backprojection and linear inverse solvers. While these assume isotropic reflectance only, it may be better to report compute and runtime complexity here instead of absolute runtime. Note that a single third-bounce transport on comparable resolution can be executed in under a second using plain CUDA on high-end GPU hardware. In the light of recent raytracing hardware acceleration, matrix-free solvers may also offer very high runtime performance, as speculated by the authors for a GPU version of their algorithm. However, as both is a bit speculative (without actually having done the GPU implementation), I would feel more comfortable softening this claim a bit in the final manuscript.

3) Comparisons: It would have been nice for reviewers to try and compare the reconstructions against other baselines, such as an isotropic solver with a quadratic program. Unfortunately, the measurements and reconstruction source code was not provided. That said, I applaud the authors for giving a very detailed description of their method in the supplemental material. To make the results fully reproducible, and facilitate future work building upon the ideas presented in this work, I'd encourage the authors to release their measurements and source code along with the publication of this manuscript.

4) Algorithm comparison overview: To parse the algorithm results a bit better, it would be great if the authors could provide a table comparing recent algorithms both in terms of compute and memory complexity. The Matlab runtimes are a bit of a strawman, as these depend on how Matlab schedules compute, organizes memory (not well), and implements array ops.

Reviewer #3 (Remarks to the Author):

The authors present a detailed account of their recent work aimed at further developing the "virtual phasor field" approach to non-line-of-sight imaging.

The phasor field approach is not itself new and has been reported before in several papers, co-authored also by the same authors of the present manuscript.

The main novelty presented here refers to adaptation of this technique with the goal of improving reconstruction quality whilst reducing the required computer effort and resources in particular memory.

I also appreciated the clear exposition of the numerical technique and underlying maths, although I do have some comments about this, as detailed below.

Given that the phasor approach to NLOS is not itself new, there is a somewhat weaker case for publishing in Nature Commun. However, I believe that the quality of the results together with the improvements and developments proposed here, will make this an important reference point for researchers not only in this field but potentially also in related fields, e.g. imaging in scattering media where similar approaches could find applications.

I would therefore suggest acceptance after some suggested changes.

Comments:

1) page 2: "To the best of our knowledge none of them have been applied to larger and more complex scenes" - I am not sure I agree with this comment. Ref. [28] in the manuscript for example, presents 3D reconstruction based on the f-K transform (i.e. different from the phasor approach) of a large, complex scene with very high quality.

2) Presentation of the virtual phasor field approach: the authors carefully present their equations together with text explaining the steps. However I feel that there are still some points that could be further clarified. Looking at Eq.s 5 and 6, it seems that the phasor fields are nothing other than the frequency components of the light pulse (either outgoing or the return signal), frequency shifted away from zero frequency by an arbitrary quantity Ω_c .

- How is this Ω_c chosen and why not just take $\Omega_c=0$, i.e. use the actual FT of the laser pulse envelope?

- The authors say, after eq. 14, that they typically deal with "dozens" of Ω values. It is not clear where this number comes from. Given that the Ω spectrum is the FT of the direct space pulse envelope, there could/should potentially be many thousands of Ω points in the spectrum, depending only on the extent of the time scale over which the temporal measurements are recorded.

- It might be useful to provide to provide a graph showing an actual example of a typical pulse shape used for eq. 5, how this transforms to eq. 6 and then how eq. 14 is applied (i.e. how the Ω spectrum is discretised before summing elements).

- I am not sure what to make of Figure 6. Maybe this was indeed an attempt to illustrate the various steps, including those mentioned in my comment. But I could not follow the logic or extract any useful information. Maybe adding the suggested graphs, thus making this a bit more quantitative, will help.

3) All of the figures lack any form of axis labels or indication of length scales, making it very hard to appreciate the results.

Ideally, an additional picture of the scene should also be included as there is no ground truth image to compare with and thus judge the actual complexity of the scene and quality of the retrieval.

4) Acquisition times are indicated throughout the paper (also figures and tables) in ms. But it is not clear if this is the acquisition time for each pixel or acquisition point or if it is the total acquisition time. In the former case, total acquisition times should be also indicated. Also, total size of camera aperture on the observation screen (relay wall) and number of scan points should be clearly indicated, for example in figure 1 or maybe in a dedicated figure showing the details of the experiment layout (and maybe also at the beginning of the Results section where the setup is described, e.g. where the detector and laser features are described).

5) What is the laser wavelength and illumination power required? Clearly, this will affect SNR so acquisition times alone do not provide a valid indication of the speed of a system that one may attempt to build in their own lab.

Reviewer 1:

The recent Nature paper on phasor fields was a valuable contribution and very interesting. Phasor fields turn time-resolved NLOS measurements into a virtual wave and then propagate that wave to the hidden scene using a free-space propagation operator, such as the Rayleigh-Sommerfeld Diffraction Integral (Eqs. 1-4). These concepts were thoroughly explored in the recent Nature paper. Unfortunately, this manuscript adds little new. To be precise, the primary contribution of this manuscript is to implement the free-space propagation operator in the Fourier domain (Eq. 8). This is common practice in computational optics and outlined in standard textbooks, such as Goodman's Fourier Optics book, and also taught in introductory Fourier optics classes.

Authors:

We thank the reviewer for their thorough review and many helpful suggestions that help improve our manuscript. R1 raises many concerns regarding statements made in our paper that we will cover below. First we would like to mention that our main contribution is to provide a fast non-approximative reconstruction method for a general non-confocal measurement. This has never ever been shown before. This core contribution does not appear to be contested by R1. We added a section (line 627 - 693) to the manuscript to better explain the inherent advantages in non-confocal NLOS capture.

*We believe there is a misunderstanding of some important details of our manuscript: R1 mentions that our convolutional solution presented in Eq. 8 is approximate, was introduced in the recent nature paper (Ref.[29]), and is described in Goodman's Book on Fourier Optics. R1 may be thinking of the Fresnel approximation. The Fresnel approximation is indeed an approximate convolutional solution to the diffraction problem that has all the asserted properties. What we however derive here is **a convolutional solution to the RSD** which is not an approximation. The reviewer is correct in that Goodman's Fourier Optics provides many fundamental insights for the diffraction imaging. The convolutional solution to the RSD integral, however does not appear to be part of it and is instead introduced as a novel contribution in recent publications (Ref.[33-35]). The fresnel approximation is a convolutional approximate solution for the diffraction propagator. It is used in the recent Nature paper and covered in Goodman's Book. As is shown in our Nature paper the Fresnel approximation has large phase artifacts and does not provide a great reconstruction even for very simple scenes. It does not work as part of the approach presented here. We are happy to add more data to show this if requested. So to emphasize again: What we are presenting here is an **exact** propagation operator directly solving the RSD. It is not an approximation. We apologize for this misunderstanding. We modified the text to better clarify.*

What is more concerning is that the authors make contradictory and false statements throughout the manuscript, possibly trying to mislead the reader and misrepresent their contributions; they also do not actually demonstrate many of their claims or quantitatively evaluate them. Therefore, I cannot support this submission.

Authors:

We apologize for the misconceptions arising from our first manuscript. Diffraction, FK Migration, and the spherical inverse radon transform (LCT) are very mature fields with large bodies of research that are unfamiliar to NLOS researchers. It is therefore difficult to summarize the important information from the related work concisely. We have tried our best to explain the matter better in the paper and address the concerns of R1 below.

One of the primary arguments that is made throughout the manuscript is that the proposed method has a lot lower memory requirements than other real-time NLOS algorithms. This is false, as the required 3D Fast Fourier Transforms in those methods can be sequentially applied to all three dimensions x,y,t to reduce their memory footprint by three orders of magnitude at the cost of increased compute times albeit with the same computational complexity of $O(N^3 \log N)$.

Authors:

We appreciate the reviewer's concern about memory complexity. The key point here is that both LCT and FK require oversampling and interpolation of the data that is the actual complexity bottleneck for memory, speed, and accuracy. The FFT itself does not contribute significantly to the memory requirements except that it requires that the entire input data and reconstructed sparse 3D volume to be stored. This alone is small compared to the re-sampling and oversampling requirements, but it is already more than what our algorithm needs.

The interpolation issue is actually stated directly in the FK Migration literature. See for example (Numerical methods of exploration seismology: with algorithms in MATLAB®. Cambridge University Press, 2019.) on the page 151:

In (k_x, k_z) space, the curves of constant f/v are semi-circles. As $k_x = 0$, $k_z = f/v$ so these hyperbolae and semi-circles intersect when the plots are superimposed.

The spectral mapping required in equation (5.53) is shown in Figure 5.49. The mapping takes a constant f slice of (k_x, f) space to a semi-circle in (k_x, k_z) space. Each point on the f slice maps at constant k_x which is directly down in the figure. It is completely equivalent to view the mapping as a flattening of the k_z hyperbolae of Figure 5.47. In this sense, it is conceptually similar to the NMO removal in the time domain though here the samples being mapped are complex valued. That the spectral mapping happens at constant k_x is a mathematical consequence of the fact that k_x is held constant while the f integral in equation (5.50) is evaluated. Conceptually, it can also be viewed as a consequence of the fact that the ERM seismogram and the migrated section must agree at $z = 0$ and $t = 0$.

On a numerical dataset, this spectral mapping is the major complexity of the Stolt algorithm. Generally, it requires an interpolation in the (k_x, f) domain since the spectral values that map to grid nodes in (k_x, k_z) space cannot be expected to come from grid nodes in (k_x, f) space. In order to achieve significant computation speed that is considered the strength of the Stolt algorithm, it turns out that the interpolation must always be approximate. This causes artifacts in the final result. This issue will be discussed in more detail in section 5.4.2.

The creation of the migrated spectrum also requires that the spectrum be scaled by $\hat{v}k_z/\sqrt{k_x^2 + k_z^2}$ as it is mapped (Equation (5.53)). In the constant velocity medium of this theory, $\sin \delta = \hat{v}k_x/f$ (δ

A naive interpolation algorithm would require computing nearest neighbors which would an N^6 computational complexity or require extra memory by storing a map of nearest neighbors computed with a more efficient algorithm. More efficient methods may be possible, but require further research. We don't consider it likely that they can be used to reduce memory needs to something comparable to our approach.

Even if we assume that re-sampling and 3D FFT can be done with no memory requirements, the methods would still require significantly more memory than what we propose due to the need to build time domain histograms and store the complete sparse 3D volumetric result of the FFT.

We will explain this in more detail in the responses below and quote some relevant statements from the associated literature. We also added a section (line 303 - 368) to the paper to clarify memory requirements further.

"The memory complexity of our algorithm is defined by the need to store the FDH and the resulting 2D image." (line 303 - 368)

In the existing published algorithms, the 3D fourier transforms do not dominate the resource need for both compute time and memory. We provide our code and the FK Migration and LCT codes for the reviewers to test with this revision. In addition to illustrate the computational and memory needs we provide a matlab script below that can be executed without any real data. We provide a matlab example code for the f-k migration method, the same method is being used for Ref.[28]. This is intended for illustrating the actual memory and run time for Stolt's method. You can run the code by simply copying text below. It will output the actual run time for each of the three steps in the f-k method. The memory profile is quite heavy which almost reaches 25GB for processing the interpolation step.

*Program output actual execution for each step: (*perform on the same PC for our results submission).*

```
f-k migration running time in each step
Step 1: Forward Fourier transform 256 x 256 x 4096 in 3.369305 seconds
Step 2: Spectral Mapping as Interpolation 256 x 256 x 4096 in 13.345769 seconds
Step 3: Inverse Fourier transform 256 x 256 x 4096 in 4.709391 seconds
>>
```

Running screenshot: The time cost for each step in f-k method

```
close all
clear
clc

%% initial parameters
N = 128; % spatial sampling
M = 2048; % temporal sampling
width = 1; % half width of the spatial sampling wall
c = 3e8; % Speed of light (meters per second)
bin_resolution = 32e-12; % Native bin resolution for SPAD is 4 ps
range = M.*c.*bin_resolution; % Maximum range for histogram

% generate the corresponding simulated spatial temporal cube
meas = randn([N, N, M]);

% convert from double to single precision
meas = single(meas);

% permute the data structure
data = permute(meas,[3 2 1]);

% Define volume representing voxel distance from wall
grid_z = repmat(linspace(0,1,M)',[1 N N]);

%% Processing f-k migration
% Define n-dim grid
[z,y,x] = ndgrid(-M:M-1,-N:N-1,-N:N-1);
z = z./M; y = y./N; x = x./N;

display(sprintf(['f-k migration running time in each step']));

% Step 0: Pad data
data = data .* grid_z.^2;
data = sqrt(data);
tdata = zeros(2.*M,2.*N,2.*N); % twice large in each dimension
tdata(1:end./2,1:end./2,1:end./2) = data;

% Step 1: FFT
tic
tdata = fftshift(fftn(tdata));
time_elapsed = toc;
display(sprintf(['Step 1: Forward Fourier transform %d x %d x %d '...
    'in %f seconds', size(tdata,3),size(tdata,2),size(tdata,1),time_elapsed]));

% Step 2: Stolt trick (Spectral Mapping)
```

```

tic
tvol = interpn(z,y,x,tdata,sqrt(abs((((N.*range)./(M.*width.*4)).^2).*(x.^2+y.^2)+z.^2)),y,x,'linear',0);
tvol = tvol.*(z > 0);
tvol = tvol.*abs(z)./max(sqrt(abs((((N.*range)./(M.*width.*4)).^2).*(x.^2+y.^2)+z.^2)),1e-6);
time_elapsed = toc;
display(sprintf(['Step 2: Spectral Mapping as Interpolation %d x %d x %d '...
'in %f seconds'], size(tvol,3),size(tvol,2),size(tvol,1),time_elapsed));

% Step 3: IFFT
tic
tvol = ifftn(fftshift(tvol));
tvol = abs(tvol).^2;
vol = abs(tvol(1:end./2,1:end./2,1:end./2));
time_elapsed = toc;
display(sprintf(['Step 3: Inverse Fourier transform %d x %d x %d '...
'in %f seconds'], size(tvol,3),size(tvol,2),size(tvol,1),time_elapsed));

```

We can also find facts about memory issues in the fk paper (Ref.[28]):

1. FK method (Ref.[28]) full resolution results in the paper are generated by the hardware platform with extensive memory:

Text from FK method (Ref.[28]) in sec 4.3 Software:

“The data are processed on a computer with 256 GB of memory and two Intel Xeon E5-2690 v4 CPUs running at 2.60 GHz. With this hardware, our unoptimized MATLAB implementations of f–k migration and the LCT take approximately 80 s and 25 s, respectively for a volume of 512^3 samples.”

2. The FK Migration method (Ref.[28]) open-source code (<https://github.com/computational-imaging/nlos-fk>) uses spatial downsampling to make the code usable on a typical PC. This is likely also why the results from the published script don't look as good as the ones in the published paper.

Code section from the released version for FK (Ref.[28]) (<https://github.com/computational-imaging/nlos-fk/blob/master/demo.m>):

```

“
% resize to low resolution to reduce memory requirements
measlr = imresize3d(meas, 128, 128, 2048); % y, x, t
tofgridlr = imresize(tofgrid, [128, 128]);
wall_size = 2; % scanned area is 2 m x 2 m
”

```

The raw dataset is captured in 512 by 512 scanning points across 2 by 2 meter relay wall. The LCT and FK re-sampling steps need extensive memory to satisfy the mathematical formula, thus it applies spatial downsampling from 512-512 to 128-128.

The proposed method is an approximation: Fourier-based free-space propagation followed by a simple max filter. Other methods, such as LCT and FK, are exact solutions to the inverse problem. Therefore, I don't see why the proposed method could be better for confocally scanned data (some of the results seem questionable, see comments below).

Authors:

*We would like to point out again that our main contributions is to provide a fast solution for the non-confocal detection scenarios where LCT and FK are no longer valid. The Rayleigh-Sommerfeld Diffraction (RSD) type of phasor field model is **not an approximation** for the time resolved non-line-of-sight application. The only approximation made is the omission of the obliquity factor. This factor accounts for angle dependent reflectance of the involved surfaces (“lambertian shading”). It is also ignored in LCT and FK Migration.*

Besides the need to approximate missing fourier coefficients by interpolation, both FK and LCT (Ref.[27,28]) are exact only for infinite size relay walls, infinite temporal band-width and infinite spatial resolution on the relay wall. For a finite size relay wall and finite time sampling, windowed fourier transforms would have to be used and their kernel may look quite similar to the RSD kernel that is derived in our method as the result of our virtual illumination pulse.

FK and LCT also do not account for noise in the signal. Our convolution kernel acts as a band pass filter only passing frequency components that actually contribute to the reconstruction. LCT and FK pass both high frequencies that are beyond the capabilities of the capture hardware and low frequencies that are not useful for the reconstruction beyond contributing noise. As a result we find that our method is more robust to noise in the photon counts and noise in the positions on the relay wall.

In addition to our main contribution which is to demonstrate an algorithm working with non-confocal data, we also show our model can also be adapted to the special case of confocal measurements where it can be compared to LCT and FK Migration.

A discussion of available photon counts is missing. It is argued that real time capture is important and that SPAD arrays could achieve that. Yet it is neither shown nor likely as these arrays have significantly worse fill factors and photon detection probabilities. The photon counts of the proposed non-confocal system should actually be similar or worse than confocally scanned data.

Authors:

Note that the fill factor is not necessarily a problem for the total achievable photon count. Small pixel sizes available in first generation SPAD arrays (e.g. MPD and Princeton Lightwave) indeed make them worse than large area single pixel SPADs when they are positioned very close to the relay wall. At large distances (>100 meter) the small pixel sizes can be compensated by the magnification of the objective optics. However we do not propose that these first generation SPAD arrays should be used. Upcoming second generation arrays have much better characteristics thanks to the use of 3D stacking allowing for the placement of processing and memory in separate layers behind the SPAD pixels. All available commercial arrays - even second generation ones - are however designed for LiDAR which has very different requirements. We therefore are designing a dedicated SPAD array for NLOS imaging. It is understandable that someone with experience using first generation SPAD arrays and without deeper knowledge in the rationale behind the design choices might come to a negative view on SPAD arrays. With the general availability of 3D stacking, however, neither fill factor, nor time resolution or maximum count rate present any unsurmountable engineering challenges.

Note also that the quantum efficiencies of SPADs are not lower than those of other sensors. Our single pixel SPAD has a quantum efficiency between 40% and 50% in the green. Typical photography cameras have similar efficiencies (<40% for Canon 5D). ICCD cameras, Photomultiplier Tubes, Streak Cameras and Photocathode based Night Vision cameras also have around 40% quantum efficiencies. The only systems we are aware of with consistently higher quantum efficiencies are back illuminated scientific CMOS and EMCCD cameras who typically exceed 90% but are very slow.

The photon counts obtained in our NLOS measurements are given in detail in (Ref.[29]). It is straightforward to see that a system collecting light from more patches on the relay wall using more SPAD pixels would collect proportionally more light and that the amount of photons collected per time unit is proportional to the number of available pixels. Achievable photon counts from an array in the same setup can be estimated given the pixel area, the number of pixels, the focal length of the objective lens, and the distance between the objective and the relay wall.

We believe that the crucial need for SPAD arrays is an important factor motivating our work and we think it is important for readers of our manuscript to understand that motivation. Therefore included a new section motivating the use of SPAD arrays in the manuscript (line 627 - 693). We thank R1 for bringing up these concerns and we believe addressing them makes our manuscript much stronger.

Finally, there seems to be some pre-processing required for Phasor Fields and this should be included in the compute time calculations / performance measurements.

Authors:

Pre-processing is required for Phasor Fields, as well as FK Migration and LCT. In all cases the Histograms H need to be generated from raw photon time stamps. The computational cost of this process depends on the number of photons and not the size of the reconstruction. This is why it is omitted in all methods. It is true that the generation of Fourier domain histograms is more time intensive than the generation of time domain histograms. This is because our current hardware measures the information in the time domain. The main reason to generate our Fourier domain histograms directly as described in section is that the time domain histograms used in other methods require a lot of memory. If we store the raw data as a time domain histogram it would be the largest structure used in our method and define our memory complexity.

If we are not concerned with optimizing memory and would like to use time domain histograms for a better runtime comparison we can also compute the Fourier domain histograms from the time domain histograms using a set of 1D FFTs (one for each captured time response). Time requirement for doing this is small compared to the rest of the algorithm. In the provided example code it takes about 1.5 seconds for the largest scene.

Examples of claims that are not demonstrated:

- The abstract motivates real-time capture with SPAD arrays, yet there is no discussion of actually capturing data with a SPAD array. Was any of the data captured in real time? Was a SPAD array used? It seems unlikely, because available SPAD arrays neither have the time resolution nor the photon detection probability (i.e., "sensitivity") to actually capture a sufficient number of third-bounce photons of diffuse room-sized scenes as claimed in the paper. No discussion of SPAD array hardware or optical setup used to capture any of the presented data is included and all of the data shown is captured, in previous work, with a single scanned SPAD.

Authors:

We apologize that R1 got the impression that we capture with an array. In our submission, we do not claim that we captured with a SPAD array. The last sentence in the abstract was intended to make this clear "... that are currently under development" (line 21). We do think it is important to realize the inherent advantages of array capture and therefore the need for algorithms that are capable of utilizing array data. One way to assess the performance of an array do this is to invert the capture geometry of the system used in Ref.[29] that is used to generate our data. This system sequentially scans 24000 laser positions with a single stationary SPAD. In each laser position the SPAD is exposed for 5 milliseconds. As is pointed out in Ref.[29] the capture path is reversible. If we used a 24000 pixel SPAD array and a single stationary laser we could collect the same data, but it would take only 5 milliseconds as all SPAD pixels can capture in parallel. We don't consider this to be a very controversial insight and believe that any reader with the necessary insight will come to a similar conclusion. In fact the existing rendering algorithms for NLOS design all render this inverted light path rather than following the actual direction of the light. We added a section (line 627 - 693) to our paper to

explain our reasoning. There are existing SPAD arrays with a sufficient number of pixels, good fill factors, and gated detection that can achieve this (Voxtel, currently a pre-release prototype, has 256 by 256 pixels with 10% fill factor, 100 ps gate, 200 ps time resolution, 16 million counts per second for all pixels combined). Since they are designed for LiDAR they are however still far from optimal for NLOS imaging. We are currently developing a SPAD array with the lab that designed our single pixel SPAD (Prof. Tosi, Polimi). Given the fast paced nature of this field we decided to not delay publication of our algorithm until an array is available.

- "SPADs can potentially be manufactured at low cost and in large arrays enabling fast parallel NLOS capture." - this statement (among others) indicates that SPAD arrays and thus real-time capture was not actually implemented, but that seemed like a major claim in the abstract and early parts of the introduction

Authors:

We present the first fast NLOS algorithm that can be used with array detectors. In addition, our algorithm is more memory efficient than other algorithms. The SPADs and arrays we are in the process of building are indeed manufactured in the same CMOS facilities and using the same methods as regular CMOS camera chips (apart from the relatively new 3D stacking). One would therefore expect that they can be offered at a similar price.

- there is no quantitative evaluation of any of the results; it is claimed that the proposed method is "better" and "less blurry" than other methods, but that is not quantified anywhere

Authors:

In the paper we only compare existing methods which could solve the non-confocal scenarios with complex $O(n^3 \log(n))$. Since we share similar roots with phasor field, we compare the fast reconstruction with the analytical integral in the Figure 7,8 (Direct Integration). Moreover, confocal solvers (LCT/FK) can be applied only with approximations (Approx. LCT and Approx. FK). We believe the images shown (Figures 7 and 8) speak for themselves. Since the ground truth geometry is not provided in any of the public dataset we use we are unsure what a meaningful quantitative analysis would be. Since the focus of our work is on computation time, non-confocal data, and memory use, we are happy to remove mentions of image quality and let the images speak for themselves. We apologize for the missing pictures for the captured scene and new figure with ground truth images are provided in our new revision. (line 385 - 398, Figure 7, 8 and Table 3)

Examples of false or misleading statement:

- "current reconstruction algorithms have computational and memory requirements that prevent real-time application on a desktop or embedded

computer" (abstract) - as discussed below, LCT and FK demonstrated NLOS imaging in real time already

Authors:

The memory requirements for LCT/FK are extensive. Real time has only been demonstrated for low resolutions and with retro-reflecting targets that yield about a 10,000 fold increase in signal. See the supporting fact above. While it is flat out impossible to capture full resolution confocal data with present systems it is possible to reconstruct in real time on a system that uses multiple high-end GPUs providing several hundred GB of graphics memory.

- "The fastest existing algorithms also require a point-scanning or confocal acquisition of the data making capture hardware inherently noisy, slow, and complex." - single SPADs have significantly better noise performance and fill factors than SPAD arrays; the complexity of scanning confocal systems is the same as commercial LiDAR systems

Authors:

Note that the NLOS capture problem is very different from LiDAR. In LiDAR and confocal microscopy, 1st bounce light is captured. All 1st bounce light returns from the illuminated pixel. It is therefore possible to direct all available light to one point and detect it with a single pixel with good light efficiency. In addition, the confocal configuration helps raise the signal above the noise and removes some multibounce components from the data. In NLOS image the light returns from the entire relay surface at once, even if only a single point is illuminated. A single pixel capture arrangement therefore misses the vast majority of the available light. NLOS capture has more in common with regular ambient light imaging. In both the signal comes from the entire observed scene at once. Like NLOS imaging, photography with a sequential pixel by pixel capture is not practical as it would take days to collect enough light (exposure time times the number of pixels).

- "Existing real-time demonstrations, therefore, use retroreflecting targets and reconstruct at resolutions far below the hardware limits." - this is false as O'Toole et al. 2018 demonstrated that their LCT algorithm achieves exactly the theoretical resolution limit

Authors:

It is correct that LCT can achieve the theoretical resolution limit. What we mean to say here is that neither LCT nor FK can approach this limit in a real time reconstruction or measurement. Capture for a full resolution reconstruction takes 10s of minutes and reconstruction of large scenes requires hundreds of GB of RAM. Real time capture for LCT and FK has only been demonstrated at greatly reduced resolutions and using retro reflective targets.

- "Running on a desktop computer, our method recovers a room-sized volume in seconds and has a computational complexity equal to the fastest existing algorithms and less memory usage." - this is not new and has been demonstrated previously

Authors:

We are not sure what the reviewer thinks is not new. The only claim of novelty we are making here pertains to the lower memory usage. As we explain above, our algorithm has much lower memory requirements than all other reconstruction methods that have been applied to more than a single object. In FK results (Ref.[28]), the person wears a retroreflective dress and the resolution is very low. Retroreflectors appear about 10,000 brighter than white diffuse surfaces in confocal scans since a retroreflector sends all laser light back to the confocal spot.

- "An algorithm suitable for fast NLOS imaging must fulfill three separate requirements: The ability to use data that can be captured in real time, a computational complexity allowing for execution in a fraction of a second..." - the proposed work does not show real-time capture and only relies on non-real-time data captured in previous work; real-time algorithms have been demonstrated in the past

Authors:

To the best of our knowledge, real time methods without the use of retroreflectors have not been demonstrated. As we explain above, SPAD arrays can be used to overcome the signal level problems of current confocal methods. We don't consider this insight particularly controversial but we do understand that a reader unfamiliar with optical systems design and SPAD electronics might not find this obvious. We have laid out our reasoning in the new section added to the manuscript. We apologize for not doing this in the first submission. Note also that even the largest scenes shown in FK (Ref.[28]) still have a significantly smaller volume than our office scene and nonetheless require a machine with 256 GB of RAM to reconstruct at full resolution.

- "Real time reconstruction of low resolution retro-reflective scenes has also been demonstrated in a confocal scanning scenario with both LCT and FK Migration methods. However, reconstruction of higher resolution scenes with diffuse surfaces is hindered by the large memory requirements and the slow confocal capture process requiring sequential point scanning capture with a single SPAD pixel." - these previous methods require a 3D FFT; this can simply be implemented by successively Fourier-transforming each of the dimensions x,y,t to reduce the memory footprint by three-orders of magnitude at the cost of slightly increased computing times (although the same order of compute time $O(N^3 \log N)$)

Authors:

Unfortunately, the 3D FFT is not the dominant memory requirement in LCT or FK. Even if the FFT, storage of the result, and resampling steps are assumed to require no memory at all, simply storing the raw data in a time domain histogram would require more memory than all memory needs of our method combined. The real memory and complexity bottleneck in both LCT and FK Migration, however, is the need for oversampling and interpolation of the data. We added a section discussing memory requirements with examples to the manuscript to clarify this point.

- "This new method performs at speed similar to LCT and FK Migration when used on confocal data, while requiring significantly less memory" - false, see comment above; it is also not clear why non-confocal measurements would be better than confocal measurements

Authors:

We added a section (line 627 - 693) explaining the signal advantages of array vs single pixel capture. It is important that this problem is very different from LiDAR in this respect. See our responses above regarding the memory requirements.

- The results in Figs. 11,12 are questionable. The reconstruction quality of LCT and FK seem a lot worse than what was presented for that same data in the FK paper.

Authors:

We use the open-source published code and data from Wave-based non-line-of-sight imaging (FK-Migration Method, Ref.[28] on the paper). The code for generating the results for LCT and FK is from here: (<https://github.com/computational-imaging/nlos-fk>). In the paper, we only use two exposures data (minimal 10mins, maximum 180mins) for the comparisons. The results we used on the paper match the same with the author's published code (Ref.[28]). Details are provided below.

As for the differences from the visual appearance, there is another missing piece on the visualization side. Lindell et.al (Ref.[28]) use the 3d volume rendering visualization for their main paper (as well as their supplementary document) without axis denoted in both dimensions. This may end up having a different (smaller/bigger) volume and might cause the differences to reproduce the results shown in their paper (Ref.[28]) by using their published code (<https://github.com/computational-imaging/nlos-fk>). We also believe that the results for the paper were rendered at higher resolution than what is provided in the official published code. Running the reconstructions at full resolution requires over 200 GB of RAM and would not be possible on most desktop computers.

Let us use the first row in Figure 11 and 12 from our paper as examples.

1st row on Figure 11 is from bike 10 mins dataset (folder: './bike/meas_10min.mat'),

1st row in Figure 12 is from the bike 180 mins dataset (folder: './bike/meas_180min.mat'). We put the results used for our submission and results generated from the published version side by side in the following. Notice that we use the same code for 2d visualization, which is provided by the FK source code (<https://github.com/computational-imaging/nlos-fk>).

For the bike 10 mins dataset (dataset folder: './bike/meas_10min.mat'):

Our submission Figure 11. LCT and FK results (last two columns from our submission)

Published code results for LCT

Published code results for FK

For the bike 180 mins dataset (dataset folder: './bike/meas_180min.mat'):

Our submission Figure 12. LCT and FK results (last two columns from our submission)

Published code results for LCT

Published code results for FK

As shown above, our submission figure 11 and figure 12 match with Ref.[28] work.

Other issues:

- non-confocal measurements are being discussed as being significantly better than confocal measurements, but it seems largely unclear why that is, because the scanning and system calibration complexity seems a lot higher for non-confocal measurements.

Authors:

We added section (line 627 - 693) clarifying the advantage of array capture in optical systems that capture multibounce or “global” illumination.

- Fig. 5 is somewhat cryptic, it's not clear what is going on

Authors:

The Figure 5 is for helping to illustrate the spatial sectioning method mentioned in the text. We are sorry for the lacking description for this figure. Additional explanation for this figure is provided in our revision document.

“... This vignetting effect is shown in Fig. 5: On the left of the figure, one distance shift B_1 is used for reconstructing both spatial regions M_1 and M_2 which is equivalent to using one section. On the right, two different distance shift values B_1 and B_2 (see Eq. (12)) are used for M_1 and M_2 ” (line 224 - 226)

In addition, we improved the figure caption:

“Figure 5: Larger field of view scenario with two versions of the office scene: virtual lens vignetting effect without spatial sectioning (left), with spatial sectioning

(right). When only one distance shift (see Eq. 12) is used for all voxels, there are large errors in voxel brightness that lead to a blurry appearance. Using two different distance shifts in different voxel regions results in crisper images. This motivates the spatial sectioning method to mimic a perfect virtual imaging system not limited to a small field-of-view.” (Figure 5’s caption)

Missing references:

- Klein et al. "Tracking objects outside the line of sight using 2D intensity images", Scientific Reports 2016 (for intensity based NLOS approaches)
- Peters et al. "Solving trigonometric moment problems for fast transient imaging", SIGGRAPH Asia 2015 (for real-time transient imaging approaches)
- Heide et al. "Non-line-of-sight imaging with partial occluders and surface normals", TOG 2019 (for occlusion-aided NLOS imaging)
- Wu et al. "Frequency analysis of transient light transport with applications in bare sensor imaging", ECCV 2012 (for iterative NLOS imaging)

Authors:

We appreciate the additional useful references. Those missing references are added in the new revision submission.

Reviewer 2:

This submission presents an exciting and timely contribution to the field of non-line-of-sight image reconstruction algorithms. Because of its convincing results, accessible technical description, and the promised dataset/source code, this work will have a high-impact and inspire other work with eventual applications across domains. The authors present a memory and compute-efficient approach based on Rayleigh-Sommerfeld Diffraction for non-confocal non-line-of-sight measurements. While the best results of previous confocal approaches are obtained by arguably "cheating" with retroreflective materials[27], that is by engineering the BRDF of the unknown hidden scene to substantially increase SNR of the confocal measurement setup, this work considers generic scenes with a non-confocal measurement arrangement. The presented reconstruction algorithm is well-motivated and explained in detail, even providing pseudo-code of the method itself. The authors could only have provided the measurement and the algorithm source code to go beyond what is provided at this stage. The method is well evaluated against recent

reconstruction methods, with only a few open points (see these listed below), and provides both a solid theoretical framework for further algorithm development in this space, as well as, a solid baseline for additional non-line-of-sight methods.

This work should be published in a timely manner, especially to keep pace with this fast-moving field. I would like the authors to consider the following requests in the final manuscript:

Authors:

We appreciate the recommendation in a timely publication manner from you.

We are also happy to see our work could set up a baseline for future research in this area. We also agree with the open points, and the changes are made, as discussed below.

1) Non-confocal acquisition: The differences between the non-confocal and confocal measurements should be highlighted early in the manuscript. I would suggest a separate paragraph/section for this comparison. Confocal measurements discard a substantial amount of measurements and this should be explicitly highlighted. I also don't quite buy the claim of prior work [27,28] that confocal detection is substantially easier to implement as gating is absolutely necessary to not suffer from massive pileup due to the direct reflection. It would significantly add to the paper if the authors describe and analyze the additional transient light transport information briefly and make compare their setup briefly to these prior works.

Authors:

We agree with the reviewer. We added a section describing in detail the signal properties in NLOS measurements and current optical limitations to highlight the potential of SPAD arrays.

This new section (line 627 - 693) mentioned the differences between the non-confocal and confocal measurements are highlighted in our new revision submission:

"... with algorithm complexity $\mathcal{O}(N^3 \log(N))$ suitable for non-confocal NLOS measurements that include detection schemes that use detector arrays as opposed to single pixel sensors. ..." (line 15 - 17)

"... The crucial limitation of these methods that we explore in more detail below is, however, that they can only utilize the light returning from the confocal location on the relay wall and thus cannot utilize the vast majority of light available in an NLOS measurement. This is illustrated in the appendix. " (line 77 - 80)

2) Memory requirements and runtime: The claims regarding memory and runtime are validated on a broad set of reconstructions. The only open item here is the comparison against matrix-free backprojection and linear inverse solvers. While these assume isotropic reflectance only, it may be better to report compute and runtime complexity here instead of absolute runtime. Note that a single third-bounce transport on comparable resolution can be executed in under a second using plain CUDA on high-end GPU hardware. In the light of recent raytracing hardware acceleration, matrix-free solvers may also offer very high runtime performance, as speculated by the authors for a GPU version of their algorithm. However, as both is a bit speculative (without actually having done the GPU implementation), I would feel more comfortable softening this claim a bit in the final manuscript.

Authors:

We appreciate the reviewer(s) concern. The current GPU implementation relies on backprojection solver, which has a higher algorithmic complexity, and the published one we are aware of runs much longer Ref.[8]. We agree that similar to the problem in the computed tomography (CT), the backprojection approach with higher complexity can be faster with a proper CUDA with high-end GPU implementation.

We soften this claim and remove the sentences below for our new revision,

~~*“... The low memory use of our method makes it a good candidate for GPU parallel implementation which may reduce reconstruction times to fractions of seconds. ...”*~~

3) Comparisons: It would have been nice for reviewers to try and compare the reconstructions against other baselines, such as an isotropic solver with a quadratic program. Unfortunately, the measurements and reconstruction source code was not provided. That said, I applaud the authors for giving a very detailed description of their method in the supplemental material. To make the results fully reproducible, and facilitate future work building upon the ideas presented in this work, I'd encourage the authors to release their measurements and source code along with the publication of this manuscript.

Authors:

We are providing the source code with this revision.

Meantime, we copy the data links for the open-source dataset we used for this work:

1. Ref.[29]:https://springernature.figshare.com/articles/Datasets_and_reconstruction_code_for_a_virtual_wave_non-line-of-sight_imaging_approach/8084987

2. Ref.[28]: <https://github.com/computational-imaging/nlos-fk>

4) **Algorithm comparison overview:** To parse the algorithm results a bit better, it would be great if the authors could provide a table comparing recent algorithms both in terms of compute and memory complexity. The Matlab runtimes are a bit of a strawman, as these depend on how Matlab schedules compute, organizes memory (not well), and implements array ops.

Authors:

We appreciate the reviewer(s)' concern.

We add more details about the algorithm complexity and memory requirement discussion in the Section "Algorithm complexity". Our method, FK Migration, and LCT all have the same memory complexity. In practice LCT and FK migration need about 100 to 200 times more memory than our method due to the need for oversampling and grid interpolation. We added an example to explain the reason behind this large memory requirement.

"The memory complexity of our algorithm is defined by the need to store the FDH and the resulting 2D image." (line 303 - 368)

The reason for us to choose Matlab is that the published code we compare against is in Matlab. We have a c++ implementation based on the OpenCV library with roughly 20MB memory usage with 6 seconds processing time. We can also make our method faster by pre-computing all the kernels used during reconstruction. This allows us to reconstruct in 0.2 seconds on a desktop computer without using GPU, but the memory is increased to about 5GB in this scenario.

Reviewer 3:

The authors present a detailed account of their recent work aimed at further developing the "virtual phasor field" approach to non-line-of-sight imaging. The phasor field approach is not itself new and has been reported before in several papers, co-authored also by the same authors of the present manuscript. The main novelty presented here refers to adaptation of this technique with the goal of improving reconstruction quality whilst reducing the required computer effort and resources in particular memory. I also appreciated the clear exposition of the numerical technique and underlying maths, although I do have some comments about this, as detailed below. Given that the phasor approach to NLOS is not itself new, there is a somewhat weaker case for publishing in Nature Commun. However, I believe that the quality of the results together with the improvements and

developments proposed here, will make this an important reference point for researchers not only in this field but potentially also in related fields, e.g. imaging in scattering media where similar approaches could find applications. I would therefore suggest acceptance after some suggested changes.

Authors:

Thank you for this recommendation.

We agree that the scope of our manuscript is less broad than the recent Nature publication. We believe however that it demonstrates a crucial capability in NLOS imaging by showing for the first time that data from array detectors can be used with a fast convolutional reconstruction algorithm. We added a section (line 627 - 693) on the importance of using array detectors in capture to highlight the relevance of this. We also believe that it will further validate the phasor approach in general which has also been used by other research groups in recent publications.

Comments:

1) page 2: "To the best of our knowledge none of them have been applied to larger and more complex scenes" - I am not sure I agree with this comment. Ref. [28] in the manuscript for example, presents 3D reconstruction based on the f-K transform (i.e. different from the phasor approach) of a large, complex scene with very high quality.

Authors:

We are sorry for the ambiguity in terms of first submission. We do not infer FK/LCT Ref.[28] can not apply to the complex scene. The paragraph containing the sentence "To the best of our knowledge ..." only refers to the work discussed in the section above. FK and LCT are discussed after. While the scenes they can reconstruct are still significantly smaller in volume than our office scene we agree that they are similar in complexity. We changed the text to avoid this misunderstanding.

To clarify the description, we adjust the text as below,

"... To the best of our knowledge, none of the works above have been applied successfully to larger and more complex scenes with the exception of the back-projection based methods. Methods that can process scenes of moderate and high volume and complexity include FK Migration, the Light Cone Transform, and Phasor-Field virtual waves which are discussed below." (line 53 - 57)

To clarify the description for confocal FK method, we also add the new sentence in the paragraph discussing current fast method (LCT and FK),

“... Both algorithms rely on 3D convolutions allowing for fast reconstruction and demonstrate the ability to recover complex scenes from confocal measurements~\cite{Ref FK}. As we will see below, both methods require interpolation over irregular 3D grids in order to approximate the data points needed for the convolutions. This requires oversampling the reconstructions and computing nearest neighbors which is associated with significant added memory requirements. The crucial limitation of these methods that we explore in more detail below is, however, that they can only utilize the light returning from the confocal location on the relay wall and thus cannot utilize the vast majority of light available in an NLOS measurement. This is illustrated in the appendix.” (line 73 - 80)

2) Presentation of the virtual phasor field approach: the authors carefully present their equations together with text explaining the steps. However I feel that there are still some points that could be further clarified. Looking at Eq.s 5 and 6, it seems that the phasor fields are nothing other than the frequency components of the light pulse (either outgoing or the return signal), frequency shifted away from zero frequency by an arbitrary quantity Ω_c .

- How is this Ω_c chosen and why not just take $\Omega_c=0$, i.e. use the actual FT of the laser pulse envelope?

Authors:

The reviewer(s) are right, and we use the frequency components of each collected temporal measurements.

The temporal measurements are collected from picosecond device with temporal resolution around 60-70 picosecond. Because of the time invariant system behavior, we can use each frequency components to calculate the outgoing field from an input illumination wavefront (field).

Overall, the reasons to choose this illumination pulse is also shown and discussed in the previous Nature Ref.[29], and the way we choose Ω_c follows the same as Ref.[29]. The Ω_c in Eq.s 5 and 6 stands for the central frequency for this virtual illumination function (or can be transformed into the central wavelength). The Ω_c central frequency (wavelength) has to be larger than twice the spatial laser or SPAD point spacing on the rely wall and larger than the time resolution of the system. In the Fourier domain the frequency of the virtual illumination $\Omega_c=0$ becomes the offset.

We add additional texts below,

“... The center frequency $\Omega_c=0$ has to be chosen according to the spatial relay wall sampling. The smallest achievable wavelength should be larger than twice the largest distance between neighboring points \vec{x}_p and \vec{x}_c and larger than the temporal resolution of the imaging hardware. For example, given a spatial sampling of 1cm, the smallest possible modulation wavelength is larger than 2cm.” (line 140 - 144)

“... the central frequency $\Omega_c=0$ as it is shown in Fig.1” (line 150)

Moreover, the frequency bandwidth (transform into wavelength) used for the reconstruction is provided in Table 4 and the relation between finite discrete sampling and wavelength is discussed under section “Discrete RSD Model and Implementation” (line 590 - 595)

We appreciate reviewer(s) suggestions, further improvements on the capture can be made to explore the limits of the existing hardware system.

- The authors say, after eq. 14, that they typically deal with "dozens" of Omega values. It is not clear where this number comes from. Given that the Omega spectrum is the FT of the direct space pulse envelope, there could/should potentially be many thousands of Omega points in the spectrum, depending only on the extent of the time scale over which the temporal measurements are recorded.

Authors:

Similar to the question above, the frequency range depends on a particular virtual illumination function. For the frequency interval used during the reconstruction, it also need to consider the discrete spatial sampling on the wall. For a typical illumination function we used for the results (such as officescene), the number of the frequency components is up to 139 (Table 4, Number of Fourier components). This is because the spectrum of this type of illumination with the Gaussian envelope is mostly concentrated along the central wavelength and otherwise almost zero. Thus based on the spacing of the scanning pattern, we can disregard most of the Fourier components in the measurements.

We agree with the reviewer(s) that the number of components also depends on the scene depth that determines the length of the recorded transients in time. We addressed this in one section in the text:

“... In large scenes, it would also increase with scene depth which would increase computational and memory complexity. To avoid this, large scenes would have to be reconstructed in multiple depth sections. In this work, we reconstruct scenes with depth up to 3 meters representing the largest complex scenes for which data

exist. For these scenes, a depth sectioning step is not necessary. ” (line 258 - 262)

In addition, we added the following text for further clarification and quantification:

“... Throughout the paper, we set γ to 0.01, meaning that all frequency components with magnitude smaller than 1% of the maximum magnitude are ignored. The discrete spacing Ω_{res} of the considered frequency components is given by the FFT frequency resolution:

$$\Omega_{res} = 2\pi f_{sampling} / N_{bins}$$

where $f_{sampling}$ is the sampling frequency of the histograms (i.e., 1/bin width) and N_{bins} the number of time bins. ” (line 251 - 257)

- It might be useful to provide to provide a graph showing an actual example of a typical pulse shape used for eq. 5, how this transforms to eq. 6 and then how eq. 14 is applied (i.e. how the Omega spectrum is discretised before summing elements).
- I am not sure what to make of Figure 6. Maybe this was indeed an attempt to illustrate the various steps, including those mentioned in my comment. But I could not follow the logic or extract any useful information. Maybe adding the suggested graphs, thus making this a bit more quantitative, will help.

Authors:

Since those two questions are related to each other, we answer them together. We are sorry for our first submission figure about the illustration of the Fourier domain histogram. We appreciate and incorporate the reviewer(s) suggestions. The new figure with additional text to better explain is shown below.

We add the virtual pulse shape as an example used in the paper in time (eq. 5) and frequency (eq. 6) domain into Fig.1,6, as shown below,

For the Fourier domain histogram (FDH), we revise our figure (Fig.6). We also add new text for better context flow,

“... We call this new capturing method a Fourier Domain Histogram (FDH) and it is shown in Fig.6. It can be written as ...” (line 277)

“... Fig.6 illustrates the generation of the FDH. Similar to the time domain histogram binning, this FDH performs binning for each captured photon.” (line 284 - 286)

3) All of the figures lack any form of axis labels or indication of length scales, making it very hard to appreciate the results. Ideally, an additional picture of the scene should also be included as there is no ground truth image to compare with and thus judge the actual complexity of the scene and quality of the retrieval.

Authors:

We add the ground truth images for the scene and a new table for describing the actual targets size and materials. We are sorry for missing the images and descriptions in our first submission. To address the length and scales, we provide a new table to illustrate the target scene depth complexity and target descriptions. We appreciate the notice from reviewer(s).

The new result figures used for our revision submission: (Figure 7 and 8)

We also include the axis indication of scale for the results in the figure caption:

*“The width of result in each dimension is 3m as details provided in Tab.4.”
(Figure 7’s caption)*

*“The width of result in each dimension is 2m as details provided in Tab.4.”
(Figure 8’s caption)*

We also add a new table for the target scene descriptions: (Table 3)

Dataset	Scene Depth (meter)	Material
Officescene 1 ms, 5 ms, 10 ms, 20 ms	0.5 m - 2.5 m	Wooden chair, white shelf, cardboard, books, plastic, white board, statue ...
Officescene 1000 ms	0.5 m - 2.5 m	Wooden chair, white shelf, cardboard, books, plastic, white board, statue ...
4	1 m	White styrofoam
44i	0.5 m - 1.3 m	White styrofoam
NLOS	0.75 m	White styrofoam
Resolution Bar	0.75 m	White styrofoam
Shelf Light On	0.8 m	white shelf, cardboard, books, plastic ...

Table 3: Target scene parameters: scene depth complexity (distance away from the relay wall), targets material.

4) Acquisition times are indicated throughout the paper (also figures and tables) in ms. But it is not clear if this is the acquisition time for each pixel or acquisition point or if it is the total acquisition time. In the former case, total acquisition times should be also indicated. Also, total size of camera aperture on the observation screen (relay wall) and number of scan points should be clearly indicated, for example in figure 1 or maybe in a dedicated figure showing the details of the experiment layout (and maybe also at the beginning of the Results section where the setup is described, e.g. where the detector and laser features are described).

Authors:

Thank reviewer(s) mentioning the missing parameters on our captured dataset, we incorporate them into our new revision submission as following.

We add total acquisition time into the description,

“ Methods comparison on office scene: Exposure time per each pixel measurement from first row to last row is 1 ms, 5 ms, 10 ms, 20 ms, 1000 ms (note that the 1000 ms Office Scene dataset was acquired with slight differences in the object location). The total acquisition time from first row to last row is 23 s, 117 s, 4 min, 8min, 390 min.” (Figure 7 caption)

“... .. Methods comparison on simple targets: Exposure time for these scenes are all 1000 ms per each pixel measurement. The total acquisition time for each target is 390 min. ” (Figure 8 caption)

We add the parameters describing the captured,

“... The experiments are performed using the non-confocal acquisition scheme. The detection aperture on the relay wall is around 1.8m by 1.3m with 1cm spacing between each captured time response. This yields 181 by 131 captured time responses for each scene. Scene target descriptions are provided in Tab.3 including scene depth complexity and target materials.” (line 394 - 398)

Also at the beginning of the “Results” section, we incorporate more details about the hardware used for the capture,

“... ... Our experimental setup consists of a gated Single-Photon Avalanche Diode (SPAD) with a Time-Correlated Single Photon Counter (TCSPC, PicoQuant HydraHarp) with a time resolution of about 30 ps and a dead time of 100 ns to measure the time response as well as a pico-second laser (Onefive Katana HP amplified diode laser with 1 W at 532 nm, and a pulse width of about 35 ps used at a repetition rate of 10 MHz) as light source.” (line 388 - 391)

5) What is the laser wavelength and illumination power required? Clearly, this will affect SNR so acquisition times alone do not provide a valid indication of the speed of a system that one may attempt to build in their own lab.

Authors:

We are sorry for the missing laser parameters used in the “Results” section. Notice that our capture hardware is the same as current NLOS in FK Ref.[28] and Ref.[29].

We added the additional text,

“... ... to measure the time response as well as a pico-second laser (Onefive Katana HP amplified diode laser with 1 W at 532 nm, and a pulse width of about 35 ps used at a repetition rate of 10 MHz) as light source. ...” (line 390 - 391)

Note that 1W of power, when scanned at high speeds <40FPS over a 1 meter relay wall would lead to an average power of about 0.1 mW/cm² which is eye safe and similar to the brightness of an I-Phone flash. In the future we believe it will be easier and more cost effective to reduce laser power and instead utilize larger SPAD arrays.

Reviewers' comments:

Reviewer #1 (Remarks to the Author):

I appreciate the authors responses to issues raised in the previous review round. After reading the revised paper, the other reviewer responses, and the point-by-point discussion, I understand that the paper presents a new, efficient computational algorithm for processing non-confocal measurements for NLOS imaging, such as would be acquired by a SPAD array. This is an improvement over existing methods which are either more computationally expensive, like in the recent Phasor Fields work, or efficient but approximate, such as was shown by Lindell et al. (2019). However, even after the revision there are inaccuracies and inconsistencies as well as missing comparisons with the manuscript which prevent me from supporting publication at this stage. I summarize each issue and discuss below.

====

"Real time" referred to in the title, abstract, and elsewhere

====

The authors state that "real-time full-resolution capture of NLOS data is feasible with emerging SPAD array detectors, but current algorithms have computational and memory requirements that prevent real-time application on a desktop or embedded computer." They further assert that their method "will enable real-time full-resolution reconstructions when used with emerging SPAD array detectors". I disagree and would challenge these assertions on two points.

(1) The authors do not demonstrate that full-resolution real-time capture of NLOS data would indeed be feasible with SPAD arrays. I still believe that even with a high-resolution SPAD array the challenge of light efficiency will preclude real-time capture as the authors describe. For example, NLOS imaging diffuse objects would still require long exposure times with SPAD arrays due to the rapid signal decay and would thus not be real time; if the authors wish to claim the opposite, they must demonstrate it with results in the paper. In my remarks below I provide further comments on some of the authors' points addressing this which they have added in Section 6.3 of the revised manuscript.

(2) While claiming that the algorithm could enable "real-time full-resolution reconstructions", the authors demonstrate their method on scenes with 150x150 spatial resolution and require several seconds (e.g. 15-20 s per Table 1) of processing per frame. So even discarding the light-efficiency argument, the claim of enabling real-time imaging at full resolution, which I read as "high-resolution", e.g. 1 megapixel, does not seem well-founded. Presumably high-resolution reconstructions with the proposed method would take several minutes and would not be real time.

So the proposed algorithm time does not seem to make significant improvements over these previously demonstrated fast imaging results for NLOS imaging in general, including confocal imaging. Specifically, Lindell et al. captured lower 32x32 resolution results at 2 fps with faster per-frame reconstruction times of ~1 s with FK (Lindell et al, 2019). Finally, O'Toole et al. recently demonstrated 5 Hz imaging with a faster GPU reconstruction at 32x32 resolution for small scenes using the LCT on a laptop (O'Toole et al. 2018, SIGGRAPH ETECH). These timings are all similar to the results presented in this paper.

Again, I see the main contribution of this paper as introducing new computation and memory efficient algorithm for processing non-confocal measurements as would be captured by SPAD arrays. The authors do not show that it could enable general real-time NLOS capture at the moderate resolution they demonstrate, much less high-resolution (e.g. 1 megapixel); however, it would enable somewhat faster reconstruction times and certainly more accurate reconstructions in non-confocal imaging scenarios.

====

Lack of quantitative analysis

====

While the authors state in the point-by-point discussion that they are unsure of what a meaningful quantitative analysis would be, they could certainly use the readily available public datasets of non-confocal simulated data to evaluate the reconstruction fidelity. One example is the dataset of Galindo et al. (2019). In these datasets, ground truth geometry is available and so can be used to benchmark reconstruction fidelity. Quantitative results should be used to augment and explain the qualitative trends displayed in the paper.

====

Complexity analysis

====

The computational complexity of the method should be reported as $N^3 \cdot M \cdot \log(N)$ rather than $N^3 \cdot \log(N)$, where M is the number of Fourier components, and N is the number of pixels along each of the three spatial dimensions in the reconstructed volume. The authors state that, "calculating the RSD reconstruction requires a 2D FFT at each of the N depth planes for each Fourier component." By my calculation, each 2D FFT requires $N^2 \cdot \log(N)$ operations, multiplied by N depth planes, multiplied by M Fourier components.

While the authors state that the number of Fourier components is "just a constant", they acknowledge in the point-by-point discussion that "the number of components also depends on the scene depth", and in the revised paper they state that, "The number of Fourier components depends on the choice of the virtual illumination pulse. In large scenes, it would increase with scene depth...".

Moreover, the number of Fourier components is comparable to the spatial resolution of the scene, and cannot be neglected. For example, the office scene is 150x150 spatial resolution with up to 139 Fourier components.

====

Memory comparison

====

I agree that the authors' method provides improvements in memory efficiency compared to FK and LCT. This seems primarily due to the capability of reconstructing a single plane at a time, whereas other methods can only reconstruct the full 3D volume at once. The authors' description of a FDH binning method that works on timestamps is also compelling.

However, the following should be addressed in the memory analysis. The authors note that the resampling step of FK contributes to its high memory consumption, but this comparison is somewhat skewed by the use of 8 ps time bins, which seems unnecessary given the system resolution of ~ 70 ps. This is also inconsistent with Table 1, which reports using what seems to be 512 time bins of 32 ps each. Additionally, Lindell et al. (2019) present FK using a similar hardware setup with 32 ps binning. After adjusting the memory analysis to use 32 ps binning, the LCT/FK methods should use roughly $150 \cdot 150 \cdot 512 \cdot 4$ bytes, which is around 46.08 MB rather than the 184.32 MB reported in the paper. The other requirements and reported memory results should be similarly adjusted to be consistent with Table 1. and Lindell et al.'s implementation on a similar hardware setup.

I also note that code included by the authors in the point-by-point discussion to run the memory analysis has inconsistencies with what is reported in the paper. The code sets the variable "bin_resolution" to 32 ps instead of 8 ps with 2048 bins, effectively calculating reconstruction

times for distances in a 20 m round trip light path, rather than the 5 m reported in the paper (L329).

====

Further Comments on Section 6.3 on SPAD array vs confocal architecture

====

The authors state that, "NLOS imaging cannot be performed efficiently with a single pixel." This is too strong of a statement and seems inaccurate. While using a SPAD array certainly eliminates the scanning requirement in measurement acquisition, the tradeoff in efficiency between SPAD arrays vs single-pixel SPAD detectors as it pertains to NLOS imaging is all about light efficiency, and both single-pixel scanned architectures and 2D sensors can be used efficiently.

Consider current state-of-the-art high-resolution SPAD arrays have pixels that are a few microns in size (e.g. the SwissSPAD2 from Ulku et al., 2019, a 512x512 sensor) compared to single-pixel detectors that have pixel sizes >100 microns (e.g. MPD sensors). Neglect fill-factor and consider that we select the optics to achieve an equal f-number where a 1 cm spot on the wall is focused onto each pixel. In this case the larger single-pixel detector would collect significantly more light than each of the smaller pixels of the SPAD array. So the single-pixel could be scanned with short exposures at each point and the large 2D array would require a single long exposure for similar total exposure times.

However, in practice high-resolution SPAD arrays come with an additional tradeoff that increases exposure time. This is because not all SPAD pixels in a 2D array contain their own time-to-digital converter. Even 3D stacking technology won't overcome this limitation because of the prohibitive bandwidth requirements for streaming out photon timestamps at megasample rates from each pixel in a high-resolution array. Instead, high-resolution SPAD arrays capture a full transient over many separate gated acquisitions. They capture photons within a few-nanoseconds-long gate, then sweep this gate in picosecond intervals, requiring many sequential acquisitions to build up the transient.

In summary, non-confocal acquisition with 2D sensors and confocal acquisition with a single-pixel SPAD each have their tradeoffs. To say unilaterally that NLOS imaging cannot be performed efficiently is inaccurate and fails to consider the nuances of each approach.

====

Presentation of results

====

Similar to other recent method, the proposed approach estimates a 3D volume. Yet, the results in figured 7,8,9,11,12 are all shown as 2D images.

From the description in the text it's not entirely clear how these 2D images are rendered. Are these maximum intensity projections or do they show the integrals along the z dimension or something else? In either case, it's actually difficult to judge the quality of these results objectively from just a single 2D perspective. The authors show show xy slices (these are included currently) but also xz and yz slices of all volume and ideally also 3D perspective renderings. Otherwise, it's impossible to say whether some reconstruction artifact from some z distance contributes to degradations of the 2D features observed on other z planes.

====

Detailed discussion and comparisons to Ahn et al.

====

After a thorough literature review, I actually found another paper that is more closely related to this submission than any other we have been discussing so far: Ahn et al. "Convolutional

Approximations to the General Non-Line-of-Sight Imaging Operator", ICCV 2019. This paper is not currently cited, but a detailed discussion and direct comparisons seem absolutely necessary.

This paper describes a geometric optics approach to non-confocal NLOS imaging. In the non-confocal setting, the current baselines used for comparison in the manuscript are the approx. LCT and approx. FK methods, but neither of these methods is actually derived for non-confocal configurations, so these comparisons seem unfair. Ahn's method on the other hand is specifically developed for non-confocal imaging; source code is available on the project website.

====

Other

====

- typo in line 402: "non-confcal" is missing an o

Reviewer #2 (Remarks to the Author):

The authors have addressed all of my remaining concerns. I do support accepting this paper. Together with the source code and experimental validation, the proposed approach is solidly evaluated. I also do not agree with the concerns from Reviewer 1. The authors present an exact convolutional model for propagation and no approximation. While the claims regarding runtime remain a bit murky, I'd argue that the LCT method was also pitched with this contribution while missing comparisons against parallelized implementations of optimization-based methods. Overall, the authors present a new propagation model that can be efficiently implemented with (relatively) low memory consumption and that provides non-approximative reconstruction results. Given the high-impact field, I'd argue that this submission should be accepted.

Reviewer #3 (Remarks to the Author):

The authors have replied to all comments in a very clear and satisfactory manner. As far as I can see, this applies to both my comments and also to those of the other referees.

I stand by my initial impression that this is a very high quality piece of work that will become a reference point for the community.

I therefore suggest acceptance and publication without any further revisions.

Reviewer 1:

I appreciate the authors responses to issues raised in the previous review round. After reading the revised paper, the other reviewer responses, and the point-by-point discussion, I understand that the paper presents a new, efficient computational algorithm for processing non-confocal measurements for NLOS imaging, such as would be acquired by a SPAD array. This is an improvement over existing methods which are either more computationally expensive, like in the recent Phasor Fields work, or efficient but approximate, such as was shown by Lindell et al. (2019). However, even after the revision there are inaccuracies and inconsistencies as well as missing comparisons with the manuscript which prevent me from supporting publication at this stage. I summarize each issue and discuss below.

Authors:

We want to thank the reviewer for their thorough review and many helpful suggestions that help improve our manuscript. We do our best to address each remaining concern below.

====

"Real time" referred to in the title, abstract, and elsewhere

====

The authors state that "real-time full-resolution capture of NLOS data is feasible with emerging SPAD array detectors, but current algorithms have computational and memory requirements that prevent real-time application on a desktop or embedded computer." They further assert that their method "will enable real-time full-resolution reconstructions when used with emerging SPAD array detectors". I disagree and would challenge these assertions on two points.

(1) The authors do not demonstrate that full-resolution real-time capture of NLOS data would indeed be feasible with SPAD arrays. I still believe that even with a high-resolution SPAD array the challenge of light efficiency will preclude real-time capture as the authors describe. For example, NLOS imaging diffuse objects would still require long exposure times with SPAD arrays due to the rapid signal decay and would thus not be real time; if the authors wish to claim the opposite, they

must demonstrate it with results in the paper. In my remarks below I provide further comments on some of the authors' points addressing this which they have added in Section 6.3 of the revised manuscript.

Authors:

We will address the comments on SPAD arrays in detail in our reply to the comments about Section 6.3, see below.

The contribution of this work is indeed the development of a fast reconstruction algorithm that can work with non-confocal data. We provide a review of the signal characteristics of NLOS imaging and multi-pixel focal plane array imaging in general in the supplement to motivate our algorithm. This section is intended to provide motivation for our work and to put it in the context of larger scale NLOS imaging efforts. We did not intend to claim it as a contribution and do not provide experimental verification of the signal dynamics of multi-pixel sensors. While we consider this demonstration an essential component of the general NLOS imaging effort, it is not the subject of this paper.

We have changed the manuscript to make this more clear: we replaced the expressions pointed out by the reviewer from previous round:

"... real-time full-resolution capture of NLOS data is feasible with emerging SPAD array detectors, but current algorithms have computational and memory requirements that prevent real-time application on a desktop or embedded computer. ..." (Text from the previous round)

, into new texts from the revised manuscript:

"... We anticipate that our method will enable real time full resolution reconstructions (i.e., only limited by the temporal resolution of the SPAD) when used with emerging SPAD array detectors that are currently under development."
(Text from the revised manuscript, line number 20-21)

We add in introductory sentence to section 6.3 to clarify that it serves as a theoretical analysis to provide motivation for our work rather than experimental demonstration of the linearity and parallelism of a focal plane array optical imaging system such as a camera or SPAD array. As we point out further below, a high resolution SPAD array, or one with closely spaced pixels (i.e. a high fill factor), is not required for NLOS imaging.

"In this section we review the fundamental constraints on NLOS capture and provide an outlook for future NLOS SPAD array sensors. This section is intended to motivate the development of the non-confocal NLOS reconstruction algorithm that is the main contribution of this paper. The section is not in itself intended as

a contribution and experimental demonstration of the signal behavior described here is subject of future work.” (line number 646-650)

(2) While claiming that the algorithm could enable "real-time full-resolution reconstructions", the authors demonstrate their method on scenes with 150x150 spatial resolution and require several seconds (e.g. 15-20 s per Table 1) of processing per frame. So even discarding the light-efficiency argument, the claim of enabling real-time imaging at full resolution, which I read as "high-resolution", e.g. 1 megapixel, does not seem well-founded. Presumably high-resolution reconstructions with the proposed method would take several minutes and would not be real time.

Authors:

Thank you for pointing out this weak point in our manuscript.

We should emphasize that by full resolution we mean a reconstruction resolution that is limited by the temporal resolution of the hardware rather than computational or signal limitations. Given the ~50 ps time resolution of current SPADs, this means that a 1-2 cm voxel grid resolution is sufficient. This maximum spatial resolution for a given time resolution is derived for example in O’Toole et. al. For our phasor field method, this also means that SPAD pixels on the relay wall should have about 1 cm side length. For a 2 m by 2 m relay wall this would mean that the largest SPAD array that could be used would be 200 by 200 pixels. In practice much less than that is needed as we describe in more detail below. Future advances in detector development (not necessarily SPAD-based) of course might lead to higher available time resolution.

Therefore, we added an additional description for the full resolution reconstructions in the abstract for our revised manuscript.

“... We anticipate that our method will enable real time full resolution reconstructions (i.e., only limited by the temporal resolution of the SPAD) when used ...” (line number 20-21)

So the proposed algorithm time does not seem to make significant improvements over these previously demonstrated fast imaging results for NLOS imaging in general, including confocal imaging. Specifically, Lindell et al. captured lower 32x32 resolution results at 2 fps with faster per-frame reconstruction times of ~1 s with FK (Lindell et al, 2019). Finally, O’Toole et al. recently demonstrated 5 Hz imaging with a faster GPU reconstruction at 32x32 resolution for small scenes using the LCT on a laptop (O’Toole et al. 2018,

SIGGRAPH ETECH). These timings are all similar to the results presented in this paper.

Authors:

The LCT/FK Migration algorithm by Lindell et. al. achieves similar reconstruction speeds to our method so reconstructions of confocal data measurements are possible in real time with either. LCT/FK, however does not work with non-confocal measurements and the challenge is with the capture time of many minutes (see above) when imaging diffuse objects. Real-time imaging at the mentioned rates can only be achieved with retroreflective targets that provide in about a 10,000 fold signal boost. Those targets are rare in reality and limit the generality of scenes that could be imaged around corners.

Therefore, we added this clarification text to the introduction:

“... Real time reconstruction of low resolution retro-reflective scenes has also been demonstrated in a confocal scanning scenario with both LCT and FK Migration methods. However, the presented confocal real time captures require retroreflective targets that return most reflected light to the moving laser/detection point, while arbitrary non-retroreflective objects require scan times of at least 10 minutes~\cite{Lindell et. al}. In this case, the bottleneck of these methods is not the computation, but the acquisition. Furthermore, reconstruction ...” (line number 86-89)

Again, I see the main contribution of this paper as introducing new computation and memory efficient algorithm for processing non-confocal measurements as would be captured by SPAD arrays. The authors do not show that it could enable general real-time NLOS capture at the moderate resolution they demonstrate, much less high-resolution (e.g. 1 megapixel); however, it would enable somewhat faster reconstruction times and certainly more accurate reconstructions in non-confocal imaging scenarios.

Authors:

We agree that the contribution of our paper is to provide a fast reconstruction algorithm for non-confocal data. We include sections about SPAD array capture as a motivation for this algorithm and not as part of the contribution of this work. We have added sentences in section 6.3 to make this more clear.

“In this section we review the fundamental constraints on NLOS capture and provide an outlook for future NLOS SPAD array sensors. This section is intended to motivate the development of the non-confocal NLOS reconstruction algorithm that is the main contribution of this paper. The section is not in itself intended as

a contribution and experimental demonstration of the signal behavior described here is subject of future work.” (line number 646-650)

====

Lack of quantitative analysis

====

While the authors state in the point-by-point discussion that they are unsure of what a meaningful quantitative analysis would be, they could certainly use the readily available public datasets of non-confocal simulated data to evaluate the reconstruction fidelity. One example is the dataset of Galindo et al. (2019). In these datasets, ground truth geometry is available and so can be used to benchmark reconstruction fidelity. Quantitative results should be used to augment and explain the qualitative trends displayed in the paper.

Authors:

Unfortunately, the public dataset provided by Galindo et. al. only provides non-confocal data with a very small number of grid points and is not suitable for our reconstructions. To provide quantitative analysis, we contacted Galindo et. al. and obtained some new rendered dataset with higher grid resolution. We included the reconstructions and quantitative analysis as a supplement in the paper as shown below.

The new texts and new results for the simulated dataset and quantitative analysis are provided in our new revised manuscript. (line number 451-457) (Figure 14)

“... Reconstructions using a rendered dataset with known ground truth are shown in Supplementary Figure 14. Our proposed method reconstructs an image of the hidden scene that resembles the image that would be captured with a camera located at the relay wall. In our reconstructions, we recover phasor field irradiance for the hidden object. It is expected that the reconstruction shows spatial distortions similar to the ones seen by a real camera, as it is shown in our supplementary materials. If an exact depth measurement is desired, these biases would have to be calibrated. This is an interesting subject for future work. ...” (line number 451-457)

(Figure 14) Additional results from simulated non-confocal datasets: Three simulated targets at 0.5 m distance from a 1 m by 1 m relay wall with a single SPAD position locates at the center. For each target, we display results as a 3D volume, a 2D front view image by choosing the maximum intensity along the depth direction and the corresponding depth error in meters. From the front view image, a 2D irradiance map of the hidden target is reconstructed. The virtual camera exhibits distortions similar to the ones seen in real cameras. Since the resulting depth error is preserved for different scenes it can likely be calibrated if more accurate depth is desired. The error appears to be consistent across the different scenes with a variation of less than one voxel.

====

Complexity analysis

====

The computational complexity of the method should be reported as $N^3 \cdot M \cdot \log(N)$ rather than $N^3 \cdot \log(N)$, where M is the number of Fourier components, and N is the number of pixels along each of the three spatial dimensions in the reconstructed volume. The authors state that, "calculating the RSD reconstruction requires a 2D FFT at each of the N depth planes for each Fourier component." By my calculation, each 2D FFT

requires $N^2 \cdot \log(N)$ operations, multiplied by N depth planes, multiplied by M Fourier components.

While the authors state that the number of Fourier components is "just a constant", they acknowledge in the point-by-point discussion that "the number of components also depends on the scene depth", and in the revised paper they state that, "The number of Fourier components depends on the choice of the virtual illumination pulse. In large scenes, it would increase with scene depth..."

Moreover, the number of Fourier components is comparable to the spatial resolution of the scene, and cannot be neglected. For example, the office scene is 150x150 spatial resolution with up to 139 Fourier components.

Authors:

We appreciate reviewer(s)' concern. We agree that the treatment of the factor M is confusing in the current version. To make it more clear, as we point out in our texts,

"... To avoid this, large scenes would have to be reconstructed in multiple depth sections. ..." (line number 266-267)

It means that we require a piecewise reconstruction of different depth sections to keep M constant for scenes with large depth.

To have a better connections between these sections, we also add sentences in our algorithm complexity section to more clarifications:

"... because the number of Fourier components is just a constant by performing reconstructions in multiple depth sections mentioned in Sec.2 ..." (line number 306-307)

====

Memory comparison

====

I agree that the authors' method provides improvements in memory efficiency compared to FK and LCT. This seems primarily due to the capability of reconstructing a single plane at a time, whereas other methods can only reconstruct the full 3D volume at once. The authors' description of a FDH binning method that works on timestamps is also compelling.

However, the following should be addressed in the memory analysis. The authors note that the resampling step of FK contributes to its high memory consumption, but this comparison is somewhat skewed by the use of 8 ps time bins, which seems unnecessary given the system resolution of ~70 ps. This is also inconsistent with Table 1, which reports using what seems to be 512 time bins of 32 ps each. Additionally, Lindell et al. (2019) present FK using a similar hardware setup with 32 ps binning. After adjusting the memory analysis to use 32 ps binning, the LCT/FK methods should use roughly $150 \times 150 \times 512 \times 4$ bytes, which is around 46.08 MB rather than the 184.32 MB reported in the paper. The other requirements and reported memory results should be similarly adjusted to be consistent with Table 1. and Lindell et al.'s implementation on a similar hardware setup.

I also note that code included by the authors in the point-by-point discussion to run the memory analysis has inconsistencies with what is reported in the paper. The code sets the variable "bin_resolution" to 32 ps instead of 8 ps with 2048 bins, effectively calculating reconstruction times for distances in a 20 m round trip light path, rather than the 5 m reported in the paper (L329).

Authors:

Thank you for your suggestions. Note that besides the ability to compute 2D slices of the result, the LCT and FK migration algorithms are memory inefficient due to the need to oversample the data in the fourier domain. They therefore need to use much larger datasets with higher resolutions to achieve similar performance. Alternatively they would have to adapt more advanced interpolation methods to obtain the fourier coefficients required in the transform. In FK migration and Radon transform literature it is typical to use at least cubic interpolation since the underlying theories require twice differentiable functions.

To make our example better match the computations done in the paper, we adapt reviewer(s)' suggestions in the memory analysis section by changing 8ps into 32ps.

"... To cover this scene setup with 32ps temporal sampling rate, at least 512 temporal sampling bins are required ...

*... For LCT/FK, one needs to store $150 * 150 * 512 * 4$ bytes which is around 46MB. ...*

*... the size of the data by 2 in each dimension by zero padding resulting in a memory need of 0.368GB ($2^3 * 46$ MB)}. This 3D dataset structure ...*

*... To store the 6 nearest neighbors of each data point requires 2.21GB ($6 \times 2^3 \times 46$ MB)}. Then the linear interpolation if implemented in this way would require 2.21GB of working memory in addition. ...
... This results in a total peak memory use of $2.21\text{GB} + 2 \times 0.368\text{GB} = 2.946\text{GB}$”
(line number 335-362)*

The reason for us to pick 8ps instead of 32ps for the memory analysis for LCT and FK is that it would improve FK reconstruction quality. LCT and FK need resampling and interpolation in spatial and temporal domain and denser sampling makes interpolation steps easier. Since LCT and FK require large memory usage, we have to perform 32ps downsampling to make it work on our lab computer (FK uses 256GB memory to create results shown in their paper).

We also want to clarify that the code included in the point-by-point discussion (not our submitted code) is not used for any results on the paper. This code in the point-by-point discussion is only used for the reviewers to understand the re-sampling and interpolation step without the context of Non-line-of-sight imaging problem.

====

Further Comments on Section 6.3 on SPAD array vs confocal architecture

====

The authors state that, "NLOS imaging cannot be performed efficiently with a single pixel." This is too strong of a statement and seems inaccurate. While using a SPAD array certainly eliminates the scanning requirement in measurement acquisition, the tradeoff in efficiency between SPAD arrays vs single-pixel SPAD detectors as it pertains to NLOS imaging is all about light efficiency, and both single-pixel scanned architectures and 2D sensors can be used efficiently.

Authors:

Unfortunately, a single pixel sensor puts fundamental limits on light efficiency since it can only utilize light coming from a small 1 cm by 1 cm patch on the relay surface. Integrating over larger areas would blur the transients in time and blur the reconstruction. There is simply a limited number of photons coming from a patch of finite area. No detector optics or sensor configuration can change that. So the only options are to increase aperture or collect light from more patches. We soften the claim regarding capture efficiency in the supplement section 6.3 and replace it with:

“... However, an NLOS imaging measurement is very different as light returns simultaneously from the entire surface of the relay wall. A single pixel detector

with high spatial resolution collects light only from a very small fraction of the relay surface at a time. ...“ (line number 661-663)

Consider current state-of-the-art high-resolution SPAD arrays have pixels that are a few microns in size (e.g. the SwissSPAD2 from Ulku et al., 2019, a 512x512 sensor) compared to single-pixel detectors that have pixel sizes >100 microns (e.g. MPD sensors). Neglect fill-factor and consider that we select the optics to achieve an equal f-number where a 1 cm spot on the wall is focused onto each pixel. In this case the larger single-pixel detector would collect significantly more light than each of the smaller pixels of the SPAD array. So the single-pixel could be scanned with short exposures at each point and the large 2D array would require a single long exposure for similar total exposure times.

Authors:

Unfortunately, larger pixel areas cannot generally be utilized to collect more light. The amount of light collected depends only on the aperture size of the objective and the area on the relay wall that is seen by the pixel. We can't collect from an area larger than about 1 cm² as that would blur our reconstruction. This is not a limitation introduced by the capture hardware, but follows directly from the data formation model. So once a 1 cm² collection area on the relay surface is achieved, a larger detector area is not helpful. If we increased the detector area beyond that point, we would be forced to reduce the focal length of the objective to reduce the magnification. If the aperture size remains the same this would reduce the f-number. We added this consideration in the text of section 6.3,

“The fundamental problem is thus that there is a limited finite number of photons per area reflecting off the relay wall. For a given aperture size, the maximum possible photon rate achieved in the measurement is thus inversely proportional to the area of relay wall used. Since the largest area a single pixel transient can be averaged over without blurring is limited (~1 cm²), one can only increase the area and thereby the photon rate by using multiple pixels collecting multiple transients simultaneously. This is entirely independent of particular technical implementations of the sensor and optics and their nuances and represents a fundamental physical constraint.” (line number 673-679)

However, in practice high-resolution SPAD arrays come with an additional tradeoff that increases exposure time. This is because not all SPAD pixels in a 2D array contain their own time-to-digital converter. Even 3D stacking technology won't overcome this limitation because of the prohibitive bandwidth

requirements for streaming out photon timestamps at megasample rates from each pixel in a high-resolution array. Instead, high-resolution SPAD arrays capture a full transient over many separate gated acquisitions. They capture photons within a few-nanoseconds-long gate, then sweep this gate in picosecond intervals, requiring many sequential acquisitions to build up the transient.

Authors:

Note that we anticipate needing far less pixels than existing commercial SPAD arrays and will not need to count more photons than typical current single and multi pixel SPAD systems. Consider this example: Since we can capture a scene in 24 seconds with one pixel it is expected that 100 pixels would capture the same number of photons in 0.24 seconds. To obtain sufficient signal, we thus need to aim 100 SPAD pixels to patches of 1 by 1 cm each on a 2m by 2m relay wall. This can be combined with scanning of the SPAD pixels and/or the laser to obtain transients from enough different points on the wall. It is thus reasonable to assume that real time capture can be achieved with a 10 by 10 pixel spad array with a fill factor of $0.01^2/2^2=0.0006$ percent. This is well within the capabilities of current SPAD technology. The total number of photons required to reconstruct a scene is about 5 to 10 million per frame. For a 5 fps readout the SPAD array this needs to process about 50 to 100 million photons per second. This is quite similar to the photon rates of current SPAD arrays.

To better explain this we added the following statement to the text in section 6.3:

“... To collect sufficient amounts of photons for real time reconstructions we anticipate needing only about 100 pixels. ...” (line number 707-708)

In summary, non-confocal acquisition with 2D sensors and confocal acquisition with a single-pixel SPAD each have their tradeoffs. To say unilaterally that NLOS imaging cannot be performed efficiently is inaccurate and fails to consider the nuances of each approach.

Authors:

While the implementation of a NLOS imaging system contains indeed a lot of nuances, the underlying constraint is actually quite simple: There is a limited finite number of photons per area reflecting off the relay wall. For a given aperture size, the maximum possible photon rate achieved in the measurement is thus inversely proportional to the area of relay wall used.

We added a clarification to the text in section 6.3 to better illustrate this central argument as stated above:

“The fundamental problem is thus that there is a limited finite number of photons per area reflecting off the relay wall. For a given aperture size, the maximum

possible photon rate achieved in the measurement is thus inversely proportional to the area of relay wall used. Since the largest area a single pixel transient can be averaged over without blurring is limited ($\sim 1 \text{ cm}^2$), one can only increase the area and thereby the photon rate by using multiple pixels collecting multiple transients simultaneously. This is entirely independent of particular technical implementations of the sensor and optics and their nuances and represents a fundamental physical constraint.” (line number 673-679)

It is however useful to look at actual hardware implementations to get a better idea of how the capture constraints play out in practice. Because of the large number of nuances that could fill a lot of pages, we just want to pick the most crucial perspectives and discuss them here.

1) Importance of fill factor/pixel sensitive area:

Of course it is desirable to have the highest possible fill factor to collect as much light as possible. However, as mentioned en passant in our comments on the first revision, the optics used together with single SPAD or SPAD array sensors play a crucial role which is most often not discussed in the literature. Let us consider a single SPAD pixel for now. The fundamental question is what the size of the SPAD observation spot on the relay wall is, and not the size of the SPAD pixel on the chip or its fill factor. The spot size on the relay wall depends on the used focusing/zoom optics and actually is the parameter of interest; any photon reflecting off this relay wall area towards the sensor will reach the sensor within its light sensitive area and will therefore be counted.

2) Discussion of SPAD array parameters

Thank you for bringing up the SwissSPAD2 work by Ulku et al. In their paper “A 512×512 SPAD Image Sensor With Integrated Gating for Widefield FLIM” (IEEE JOURNAL OF SELECTED TOPICS IN QUANTUM ELECTRONICS, VOL. 25, NO. 1), they describe it in more detail. This sensor has more pixels than the largest array useful for NLOS imaging (as stated above we can use at most 200 by 200 and would be okay with 10 by 10).

As stated by the reviewer, it is true that in existing SPAD arrays, not all pixels have their own time-to-digital converter. This, however, is not a problem at all: NLOS imaging in general deals with low photon counts (which by the way is the main reason for the desire to increase the fill factor)! In [33], the photon counts for the above scenario also have been published (Extended Data Table 1). For 10 ms acquisition time, on average, 0.14 photons per time bin have been registered, while the maximum is 18 in one bin.

We don't think it is constructive at this point to discuss all the details of existing SPAD array sensors and their feasibility for NLOS imaging, because these are not specifically designed for this use. However, we want to point out that SPAD array technology is moving very fast (also because of other applications such as depth and fluorescence imaging), and the current parameters are in the ballpark of NLOS imaging. It might be necessary to custom design a SPAD array for NLOS imaging and we are doing exactly that in collaboration with our partners.

However, as stated in our previous response, “Given the fast paced nature of this field we decided to not delay publication of our algorithm until an array is available”.

====

Presentation of results

====

Similar to other recent method, the proposed approach estimates a 3D volume. Yet, the results in figured 7,8,9,11,12 are all shown as 2D images.

From the description in the text it's not entirely clear how these 2D images are rendered. Are these maximum intensity projections or do they show the integrals along the z dimension or something else? In either case, it's actually difficult to judge the quality of these results objectively from just a single 2D perspective. The authors show show xy slices (these are included currently) but also xz and yz slices of all volume and ideally also 3D perspective renderings. Otherwise, it's impossible to say whether some reconstruction artifact from some z distance contributes to degradations of the 2D features observed on other z planes.

Authors:

Thank you for your suggestions and concerns.

Volume renderings, such as the ones used in ~\cite{Lindell et. al, FK Migration} and ~\cite{Liu et. al, phasor field} are obtained by 2D projections of the data that require filtering and thresholding similar in effect to the max filter we use in our method. Since ~\cite{Lindell et. al, FK Migration} do not disclose the details of the filter they use, we can only reproduce the visualization done in ~\cite{Liu et. al, phasor field}.

We follow the reviewer(s)' suggestions for 3D volume display and add more clarifications on the visualization method used for 2D images for our revised manuscript. Please see the descriptions below.

We added this visualization from reviewer(s)' suggestions for the reconstructed scene in Figure 15 (attached below).

(Figure 15) Three-dimensional volume rendering of 20ms officescene

The visualization method used for Figure 7,8,9,11,12 is also clarified in our new revised manuscript,

“... Reconstructions with the maximum intensity projection along depth direction are shown in Fig.7 and Fig.8. Results with the three dimensional volume rendering are in our supplementary materials. ...}” (line number 405-407)

====

Detailed discussion and comparisons to Ahn et al.

====

After a thorough literature review, I actually found another paper that is more closely related to this submission than any other we have been discussing so far: Ahn et al. "Convolutional Approximations to the General Non-Line-of-Sight Imaging Operator", ICCV 2019. This paper is not currently cited, but a detailed discussion and direct comparisons seem absolutely necessary.

This paper describes a geometric optics approach to non-confocal NLOS imaging. In the non-confocal setting, the current baselines used for comparison in the manuscript are the approx. LCT and approx. FK methods, but neither of these methods is actually derived for non-confocal configurations, so

these comparisons seem unfair. Ahn's method on the other hand is specifically developed for non-confocal imaging; source code is available on the project website.

Authors:

We appreciate your concerns.

First, we add Ahn et al. ICCV 2019 to our related work and reference list.

“... analysis of missing features based on time-resolved NLOS measurements, convolutional approximations to incorporate priors into filtered backprojection~\cite{Ahn et al. ICCV 2019}, occlusion-aided NLOS imaging using SPADs ...” (line number 50-51)

Second, Ahn et al. (ICCV 2019) only provide an approximate solution. While this method enables the incorporation of regularizers, it builds on filtered backprojection (Velten et al. 2012 Nature Communication) and therefore is limited by it and its complexity N^5 (this is explained in their paper Ahn et al. (ICCV 2019) convolution complexity section).

“We can break down the cost of solving problems (P2) and (P3) into two parts. The first part is using backprojection to compute the backprojected volumetric albedo ...” (texts from 4.2 computational complexity Ahn et al. (ICCV 2019))

Because of the huge time requirement of backprojection and the fact that Ahn’s method is an approximation, we do not further consider it in the context of our manuscript.

====

Other

====

- typo in line 402: "non-confcal" is missing an o

Authors:

Thank you for pointing out the typo. It is corrected in our revised manuscript.

Reviewer 2:

The authors have addressed all of my remaining concerns. I do support accepting this paper. Together with the source code and experimental validation, the proposed approach is solidly evaluated. I also do not agree with the concerns from Reviewer 1. The authors present an exact convolutional model for propagation and no approximation. While the claims regarding runtime remain a bit murky, I'd argue that the LCT method was

also pitched with this contribution while missing comparisons against parallelized implementations of optimization-based methods. Overall, the authors present a new propagation model that can be efficiently implemented with (relatively) low memory consumption and that provides non-approximative reconstruction results. Given the high-impact field, I'd argue that this submission should be accepted.

Authors:

We appreciate your recommendation for accepting our work.

Reviewer 3:

The authors have replied to all comments in a very clear and satisfactory manner. As far as I can see, this applies to both my comments and also to those of the other referees. I stand by my initial impression that this is a very high quality piece of work that will become a reference point for the community. I therefore suggest acceptance and publication without any further revisions.

Authors:

Thank you for your recommendation for publication without any further revisions. We appreciate your suggestions to our paper.

REVIEWERS' COMMENTS:

Reviewer #1 (Remarks to the Author):

I appreciate the authors' detailed response and edits they have made to the manuscript. Most of my concerns have been addressed in a satisfactory manner. Although I do not fully agree with all the statements in the response or manuscript, I will not further delay publication of this manuscript, especially given the enthusiasm of the other reviewers.

Here are a few more optional comments and suggestions that the authors could consider for the camera-ready version (but I hope they will not lead to another review cycle):

- l. 8: "While real time full resolution capture of NLOS data is feasible with emerging Single-Photon avalanche diode (SPAD) array detectors" -> see comments from the last round; this is speculative and neither demonstrated in this paper nor anywhere else; I remain skeptical if it is possible at all in the near future

- l. 20: "We anticipate that our method will enable real time full resolution reconstructions (i.e., only limited by the temporal resolution of the SPAD)" -> again this is speculative and the phrase "full resolution" remains ambiguous

- quantitative experiments in supplement: it seems odd that data rendered with a computer has any kind of optical aberrations / distortions, most computer graphics techniques do not have that and captured data could easily be undistorted using conventional camera calibration techniques. the discussion of this effect does not seem convincing. it may also be helpful if the authors reported RMSE error of the entire reconstructed scenes (the error map is good, but having another number would be good). also, the units on the colorbar of figure 14 are missing.

- the 3D rendering included in Figure 15 does not add anything to the manuscript. as mentioned before, it would be good to see top-down views or other perspectives that help the reader adequately assess the quality of the results. for example, fig. 14 seems to have some distortion in z, which would be easier to see from the top. similarly, other results such as the ones in figures 7-9, 11, 12, would also benefit from top down views or other perspectives. Just looking at the max intensity projection in xy can be misleading and only shows lateral, but not axial resolution achieved with these methods

- the discussion of Ahn's method seems limited, given that it solves the same non-confocal NLOS problem as the proposed method

Reviewer 1:

I appreciate the authors' detailed response and edits they have made to the manuscript. Most of my concerns have been addressed in a satisfactory manner. Although I do not fully agree with all the statements in the response or manuscript, I will not further delay publication of this manuscript, especially given the enthusiasm of the other reviewers. Here are a few more optional comments and suggestions that the authors could consider for the camera-ready version (but I hope they will not lead to another review cycle):

Authors:

We want to thank all reviewer for their thorough review and many helpful suggestions that help improve our manuscript.

- I. 8: "While real time full resolution capture of NLOS data is feasible with emerging Single-Photon avalanche diode (SPAD) array detectors" -> see comments from the last round; this is speculative and neither demonstrated in this paper nor anywhere else; I remain skeptical if it is possible at all in the near future

Authors:

We agree that the outlook on SPAD sensors we provide is not currently demonstrated in direct experiments and is intended to motivate the presented algorithm. It is not essential for us to make specific predictions about SPAD arrays sensors and their potential as we are planning to provide a demonstration of the capabilities of SPAD arrays in upcoming projects. We would also realize that the focus on SPAD arrays may be a distraction from our key argument which is about the advantage of non-confocal multi-pixel capture over confocal single-pixel capture. There are indeed many different ways of realizing a non-confocal measurement that captures light from the entire relay surface.

We therefore further soften the statements regarding SPAD in our final manuscripts:

"... with a computational inverse method. While capture systems capable of collecting signal from the entire NLOS relay surface can be much more light efficient than single pixel point scanning detection, current reconstruction algorithms ..."

- I. 20: "We anticipate that our method will enable real time full resolution reconstructions (i.e., only limited by the temporal resolution of the SPAD)" -> again this is speculative and the phrase "full resolution" remains ambiguous

Authors:

The resolution of a NLOS reconstruction is limited by the time resolution of the detection system (see e.g. O'Toole 2018). For a SPAD, time resolution is 30 ps at best leading to a theoretically achievable grid resolution of 1 cm in the hidden scene.

We therefore further revise our final manuscripts:

"... mentioned above. In addition, the resolution of an NLOS reconstruction is limited by the time resolution of the detection system~\cite{OToole_18}. For a SPAD, the time resolution is 30ps at best leading to a theoretically achievable grid resolution of 1cm in the hidden scene. Methods that can process ..."

"... We anticipate that our method will enable real time reconstructions with resolutions only limited by the temporal resolution of the sensor when used with emerging SPAD array detectors. ..."

- quantitative experiments in supplement: it seems odd that data rendered with a computer has any kind of optical aberrations / distortions, most computer graphics techniques do not have that and captured data could easily be undistorted using conventional camera calibration techniques. the discussion of this effect does not seem convincing. it may also be helpful if the authors reported RMSE error of the entire reconstructed scenes (the error map is good, but having another number would be good). also, the units on the colorbar of figure 14 are missing.

Authors:

We appreciate the reviewer's concerns and suggestions. Aberration correction is an interesting subject for future research. If accurate depth is desired, it is also advisable to use the phasor field phase which would be more accurate than the magnitude reconstructions presented here. We will explore high precision depth reconstructions using phasor field interferometry in future research.

We incorporate the reviewer's suggestion to include the RMSE error for the simulated results as shown in the caption of Figure 14 (new: Supplementary Figure 2). We compute the RMSE error for pixels that have a depth in the depth map. As for the Figure 14 the units are given in the captions. We believe this is in agreement with Nature Communications formatting guidelines as well.

Therefore, we incorporate your suggestions for figure 14 and its caption in our final manuscripts as follows:

"... less than one voxel. The root-mean-square error values for three simulated targets are 0.0097 m, 0.0178 m and 0.0257 m, respectively."

- the 3D rendering included in Figure 15 does not add anything to the manuscript. as mentioned before, it would be good to see top-down views or other perspectives that help the reader adequately assess the quality of the results. for example, fig. 14 seems to have some distortion in z, which

would be easier to see from the top. similarly, other results such as the ones in figures 7-9, 11, 12, would also benefit from top down views or other perspectives. Just looking at the max intensity projection in xy can be misleading and only shows lateral, but not axial resolution achieved with these methods

Authors:

Thank you for the suggestions for Figure 15. We incorporate the reviewer's suggestion by adding all three perspectives (front view, top view, side view) and volume renderings to our new manuscript as follows:

Revised Figure 15:

New Figure 15 (new: Supplementary Figure 3) caption:

“... shows the captured geometry. **b-d.** show the reconstructed image by the maximum intensity projection from front, top and side views. **e.** shows three dimensional volume of the reconstruction.”

- the discussion of Ahn's method seems limited, given that it solves the same non-confocal NLOS problem as the proposed method

Authors:

We appreciate the reviewer's suggestion for an additional discussion.

Recent work by Ahn et. al. 19 uses priors to improve the reconstruction quality by using an iterative steps. Since the method involves a backprojection step it has the same complexity and performance limitations as a standard back-projection based method.

Therefore, we add an additional discussion of Ahn's method in our final manuscript as follows:

“... with the exception of the back-projection based methods. Ahn et al.~\cite{Ahn 19} can improve the reconstruction quality after the back-projection via an iterative convolution step. Since the method involved a back-projection as it's first step it shares the speed and complexity disadvantages of the back-projection based methods mentioned above. Methods that can process scenes ...”